# Modelling TFE renal cell carcinoma in mice reveals a critical role of WNT signaling

Alessia Calcagnì[1], Lotte kors[1,2], Eric Verschuren[3,4], Rossella De Cegli[1], Nicolina Zampelli[1], Edoardo Nusco[1], Stefano Confalonieri[5,6], Giovanni Bertalot[5], Salvatore Pece[5,7], Carmine Settembre[1,8,9,10,11], Gabriel G Malouf[12,13,14,15], Jaklien C Leemans[2], Emile de Heer[3], Marco Salvatore[16], Dorien JM Peters[4], Pier Paolo Di Fiore[5,6,7], Andrea Ballabio[1,8,9,10,11]*

[1]Telethon Institute of Genetics and Medicine, TIGEM, Pozzuoli, Naples, Italy; [2]Department of Pathology, Academical Medical Center, Amsterdam, The Netherlands; [3]Department of Pathology, Leiden University Medical Center, Leiden, The Netherlands; [4]Department of Human Genetics, Leiden University Medical Center, Leiden, Netherlands; [5]Molecular Medicine Program, European Institute of Oncology, Milan, Italy; [6]IFOM, The FIRC Institute for Molecular Oncology Foundation, Milan, Italy; [7]Department of Oncology and Hemato-Oncology, University of Milan, Milan, Italy; [8]Department of Molecular and Human Genetics, Baylor College of Medicine, Houston, United States; [9]Jan and Dan Duncan Neurological Research Institute, Texas Children Hospital, Houston, United States; [10]Medical Genetics, Federico II University, Naples, Italy; [11]Medical Genetics, Department of Medical and Translational Sciences, Federico II University, Naples, Italy; [12]Department of Medical Oncology Groupe Hospitalier Pitie-Salpetriere, University Paris 6, Paris, France; [13]Assistance Publique Hopitaux de Paris, University Paris 6, Paris, France; [14]Faculty of Medicine Pierre et Marie Curie, University Paris 6, Paris, France; [15]Institut Universitaire de Cancerologie GRC5, University Paris 6, Paris, France; [16]IRCCS-SDN, Naples, Italy

*For correspondence: ballabio@tigem.it

Competing interests: The authors declare that no competing interests exist.

**Abstract** *TFE*-fusion renal cell carcinomas (*TFE*-fusion *RCCs*) are caused by chromosomal translocations that lead to overexpression of the *TFEB* and *TFE3* genes (*Kauffman et al., 2014*). The mechanisms leading to kidney tumor development remain uncharacterized and effective therapies are yet to be identified. Hence, the need to model these diseases in an experimental animal system (*Kauffman et al., 2014*). Here, we show that kidney-specific *TFEB* overexpression in transgenic mice, resulted in renal clear cells, multi-layered basement membranes, severe cystic pathology, and ultimately papillary carcinomas with hepatic metastases. These features closely recapitulate those observed in both *TFEB*- and *TFE3*-mediated human kidney tumors. Analysis of kidney samples revealed transcriptional induction and enhanced signaling of the WNT β-catenin pathway. WNT signaling inhibitors normalized the proliferation rate of primary kidney cells and significantly rescued the disease phenotype in vivo. These data shed new light on the mechanisms underlying *TFE*-fusion *RCCs* and suggest a possible therapeutic strategy based on the inhibition of the WNT pathway.

## Introduction

The MIT/TFE family of bHLH leucine zipper transcription factors includes the *MITF, TFEB, TFE3* and *TFEC* genes, which are master regulators of cell homeostasis, growth and differentiation

(*Levy et al., 2006*; *Sardiello et al., 2009*; *Settembre et al., 2011*). All family members are able to both homodimerize and heterodimerize with each other through their bHLH-LZ domain (*Hemesath et al., 1994*). These transcription factors bind a DNA sequence called the M-box and a non-canonical E-box sequence (TCATGTG, CATGTGA or TCATGTGA) (*Hemesath et al., 1994*; *Aksan and Goding, 1998*). A large body of evidence indicate that they play an important role in many cellular and developmental processes.

*TFEB* was found to regulate a large gene network, named *Coordinated Lysosomal Expression and Regulation (CLEAR)*. This network includes many genes involved in lysosomal biogenesis and autophagy (*Sardiello et al., 2009*; *Palmieri et al., 2011*). Several studies have shown that TFEB responds to a variety of stimuli and stress conditions, such as starvation, and acts as a master regulator of the lysosomal-autophagic pathway and of cellular clearance (*Ballabio, 2016*; *Roczniak-Ferguson et al., 2012*; *Sardiello et al., 2009*; *Settembre et al., 2011*, *2012*; *Settembre and Medina, 2015*; *Martina et al., 2014b*). Recent data indicate that the *TFEB* and *TFE3* genes regulate a similar set of genes and have partially redundant function (*Martina et al., 2014a*).

Renal cell carcinomas originate from the renal epithelium and include several subgroups defined according to their histological phenotype. The most frequent RCCs are papillary (15–20%), Clear Cells (65–70%) and cromophobe (5–10%) (*Amin et al., 2002*). In these categories, mutations in 12 different genes (*VHL, MET, FH, FLCN, SDHB, SDHC, SDHD, TSC1, TSC2, PTEN, MITF* and *BAP1*) have been associated with an increased susceptibility of developing RCC (*Linehan and Ricketts, 2013*). *TFE*-RCCs are a group of renal cell carcinomas caused by chromosomal translocations involving *TFEB* and *TFE3* genes (*Kauffman et al., 2014*) and representing around 2% of all RCCs (*Komai et al., 2009*), and almost 12% of papillary type II RCCs (*Linehan et al., 2015*).

Recent TCGA analyses revealed that the gene fusions caused by chromosomal translocations involving *TFEB* and *TFE3* are the only recurrent translocations in the kidney (*Linehan et al., 2015*; *Malouf et al., 2014*). In the case of TFEB, a recurrent chromosomal translocation t(6;11) (p21;q13) involves the promoter of the non-coding Alpha gene and the transcription factor EB (*TFEB*) (*Argani et al., 2001*, *2005*). As a consequence, *TFEB* falls under the control of the strong Alpha gene promoter, resulting in a high (up to 60-fold) overexpression of a structurally normal TFEB protein (*Kuiper et al., 2003*). More recently, additional TFEB translocation partners were described, such as the *KHDBRS2* (inv(6) (p21;q11)) (*Malouf et al., 2014*) and the *CLTC* (t(6;17) (p21;q23)) (*Durinck et al., 2015*) genes. Tipically, these tumors show nests of epithelioid cells with clear cytoplasm, known as clear cells (CCs), and clusters of small cells, usually around the multi-layered basement membrane (mBM) made up of hyaline material (*Argani et al., 2005*). Some cases presented with areas of a tubular or cystic structure covered by a single layer of flattened cuboidal to columnar cells with clear cytoplasm, mimicking clear cell RCC with cystic changes (*Rao et al., 2012*). Currently, *TFEB* translocation, overexpression and nuclear localization are considered as a diagnostic marker for the disease. Initially, these tumors were mainly observed in pediatric patients, but now they are considered relatively common in young adults (*Komai et al., 2009*). The mechanisms leading from *TFE3/TFEB* gene overexpression to kidney tumor development remain largely uncharacterized, thus the need for modeling these diseases in experimental animal systems for the identification of effective targeted therapies.

Here, we show the generation and characterization of two different transgenic mouse lines that overexpress TFEB specifically in the kidney in a constitutive and inducible manner, respectively, which recapitulate both the cystic changes and the cancer phenotype of the human pathology. An extensive molecular and biochemical characterization of kidneys, as well as of primary kidney cells, derived from these mice revealed a significant hyper-activation of the WNT pathway, suggesting that this signalling pathway plays an important role in TFEB-driven kidney cancer. Finally, the use of small molecules able to specifically inhibit the WNT pathway resulted in a significant rescue of both the cystic and cancer phenotypes. These data may open the way to a new therapeutic strategy for this type of tumors.

## Results

### Generation of the transgenic mouse lines

To study the mechanisms underlying tumor development in *TFEB*-fusion *RCC*, we generated a transgenic mice that specifically overexpress *TFEB* in the kidney. We crossed a previously generated *Tfeb* conditional overexpressing mouse line that carries *Tfeb-3xFlag*$^{fs/fs}$ under the control of a strong chicken beta-actin (CAG) promoter (*Settembre et al., 2011*), herein referred to as *Tfeb*$^{fs/fs}$, with the *Cdh16*$^{Cre}$ (*Cadherin16*$^{Cre}$) mouse line, in which the *Cre* recombinase is specifically expressed in renal tubular epithelial cells starting from embryonic stage E12.5 (*Shao et al., 2002*).

In addition, to assess the effects of *Tfeb* overexpression during kidney development, we generated a second transgenic line by crossing the *Tfeb*$^{fs/fs}$ mice with a mouse line that carries a tamoxifen-inducible *CreErt2* element under the control of a *Cdh16* promoter (*Cdh16*$^{CreErt2}$promoter) (*Lantinga-van Leeuwen et al., 2006*) (*Figure 1—figure supplement 1A*). *Cdh16*$^{Cre}$*::Tfeb*$^{fs}$ and *Cdh16*$^{CreErt2}$*::Tfeb*$^{fs}$ double heterozygous mice were generated from these crossings (*Figure 1—figure supplement 1B and C*). We checked both the constitutive and inducible lines for renal *Tfeb* overexpression and confirmed that *Tfeb* mRNA levels were highly increased, and further increasing with time (*Figure 1—figure supplement 1D*). Consistently, immunoblot experiments revealed increased levels of Tfeb-3xFLAG protein in kidneys from *Cdh16*$^{Cre}$*::Tfeb*$^{fs}$ and *Cdh16*$^{CreErt2}$*:: Tfeb*$^{fs}$ mice (*Figure 1—figure supplement 1E*).

### Progressive cystic pathology in transgenic mouse lines

At sacrifice, kidneys from adult *Cdh16*$^{Cre}$*::Tfeb*$^{fs}$ and tamoxifen-treated *Cdh16*$^{CreErt2}$*::Tfeb*$^{fs}$ mice completely filled the abdominal cavity (*Figure 1A*). An increase in kidney size from *Cdh16*$^{Cre}$*:: Tfeb*$^{fs}$ mice was observed starting at P12, with a sensible increase in size detected at P30 (*Figure 1B*). A striking increase in the Kidney to Body Weight (KW/BW) ratio was also observed at this stage (*Figure 1C*). A severe enlargement of the kidneys and a significant increase in the Kidney to Body Weight (KW/BW) ratio were also observed in *Cdh16*$^{CreErt2}$*::Tfeb*$^{fs}$ mice induced with tamoxifen at several developmental stages (P12, P14, P30) (*Figure 1—figure supplement 2A and B*). These abnormalities were less severe in mice induced at P30 (*Figure 1—figure supplement 2B*). Survival time of *Cdh16*$^{Cre}$*::Tfeb*$^{fs}$ mice was approximately 3 months (*Figure 1D*). Interestingly, a late induction of *Tfeb* overexpression in *Cdh16*$^{CreErt2}$*::Tfeb*$^{fs}$ mice resulted in a slower development of the phenotype, with less severe kidney enlargement and overall increase in the survival rate (*Figure 1D*). Renal function from *Cdh16*$^{Cre}$*::Tfeb*$^{fs}$ and *Cdh16*$^{CreErt2}$*::Tfeb*$^{fs}$ mice was severely affected, as observed by the strong increase in blood urea and albuminuria (*Figure 1—figure supplement 2C*). High-frequency ultrasound and histological analysis of kidneys from both *Cdh16*$^{Cre}$*:: Tfeb*$^{fs}$ and *Cdh16*$^{CreErt2}$*::Tfeb*$^{fs}$ mice revealed the presence of a severe cystic disease (*Figure 1E*, *Figure 1—figure supplement 2D and E*). In *Cdh16*$^{Cre}$*::Tfeb*$^{fs}$ mice, small cysts arose mainly from the cortex and outer medulla at P12 and became significantly enlarged at P30. At P90, kidney architecture was completely disrupted by cysts (*Figure 1F*). *Cdh16*$^{CreErt2}$*::Tfeb*$^{fs}$ mice induced at P12 with tamoxifen and sacrificed at P90 showed a higher number of smaller cysts in both cortex and outer medulla (*Figure 1F*). Cysts were also observed in *Cdh16*$^{CreErt2}$*::Tfeb*$^{fs}$ induced at P14 and, to a lesser extent, at P30 (*Figure 1—figure supplement 2E*). Tubular epithelial cells lining the cysts showed high levels of cadherin 16, indicating the presence of *Cdh16*$^{Cre}$-mediated *Tfeb* overexpression in these cells (*Figure 1G*). Histological analysis revealed that cysts from *Cdh16*$^{Cre}$*::Tfeb*$^{fs}$ mice were positive for AQP2 and THP and negative for megalin, indicating that they originate from collecting ducts and distal tubules and not from proximal tubules. Notably, the largest cysts were almost completely negative to all tubular markers, suggesting that they became undifferentiated. Conversely, cysts from *Cdh16*$^{CreErt2}$*::Tfeb*$^{fs}$ mice were positive to megalin and THP, indicating that they arose from proximal and distal tubules (*Figure 1H*, *Figure 1—figure supplement 3*). These differences in cyst origin have already been described in other polycystic kidney disease mouse models and have been attributed to intrinsic differences of specific renal segments at different developmental stages (*Lantinga-van Leeuwen et al., 2007*; *Happé et al., 2009*; *Leonhard et al., 2016*; *Piontek et al., 2007*).

Cysts were lined by either a single layer-flattened cuboidal epithelium (sCy), or by a multilayer epithelium (mCy), indicating a de-regulation of tubular cell proliferation (*Figure 1I*). We also noticed

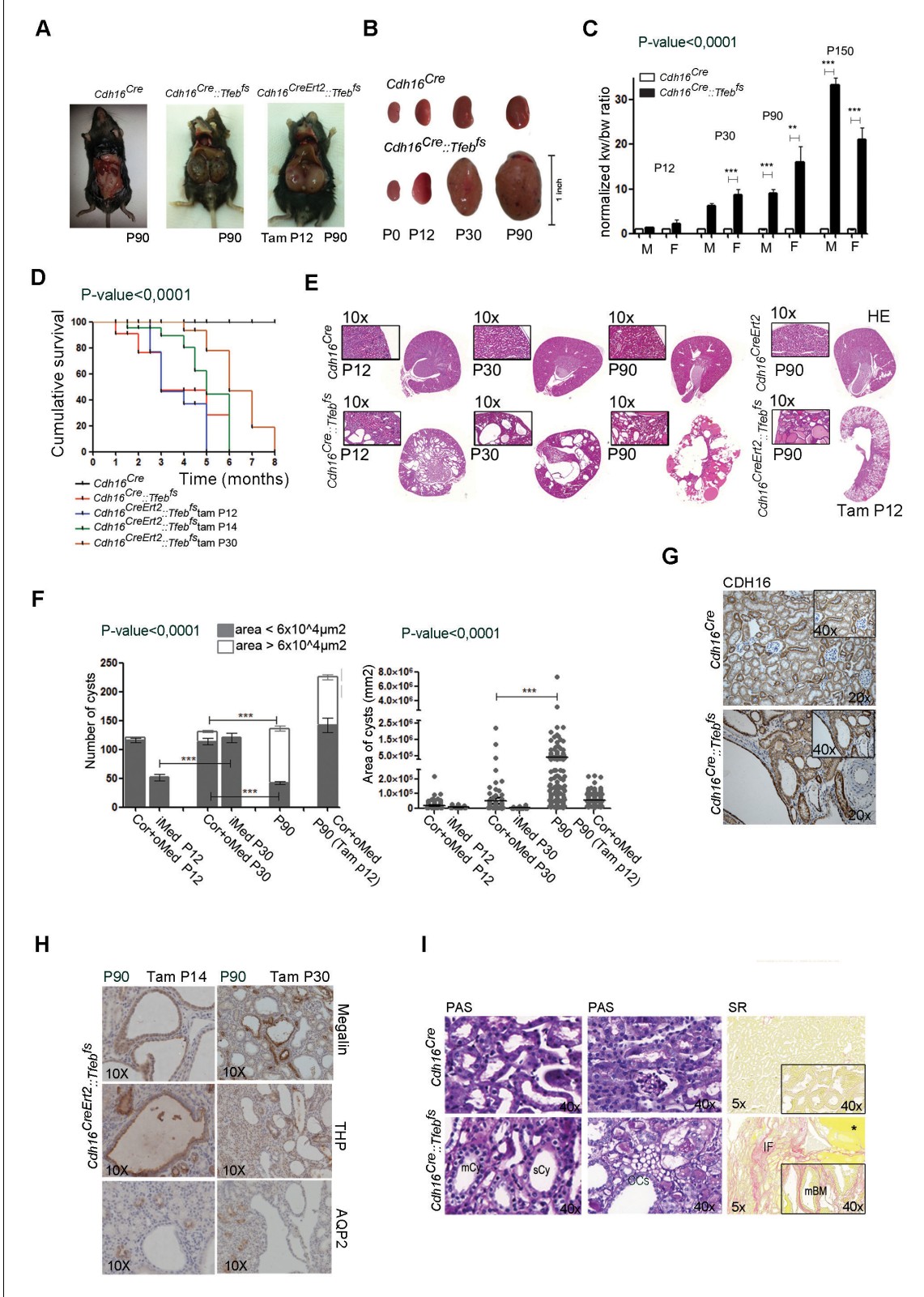

**Figure 1.** *Tfeb* overexpressing mice display cystic kidneys. Morphological analyses were performed on *Cdh16^Cre^* and *Cdh16^Cre^::Tfeb^fs^*, and on tam-treated *Cdh16^CreErt2^* and *Cdh16^CreErt2^::Tfeb^fs^* mice. (**A**) Representative images of the abdominal cavity at P90. (**B**) Kidney size at different stages (p=days post-natal). (**C**) Relative ratio of kidney-to-body weight (KW/BW). Data from males (M) and females (F) are shown separately as means of *Cdh16^Cre^:: Tfeb^fs^* to *Cdh16^Cre^* KW/BW ratio. Three-way Anova was applied (factors: gender, time, genotype). (**D**) Evaluation of the survival of *Cdh16^Cre^::Tfeb^fs^* and

*Figure 1 continued on next page*

*Figure 1 continued*

tam-treated *Cdh16^CreErt2^::Tfeb^fs^* mice. Mantel-Cox test was applied (*Cdh16^CreErt2^::Tfeb^fs^* tam P12/tam P14 p-value 0.02; *Cdh16^CreErt2^::Tfeb^fs^* tam P12/P30 p-value<0.0001). (**E**) Haematoxylin and Eosin (HE) staining of kidneys. Enlarged panels show cyst growth over time. (**F**) Number (left graph) and area (right graph) of kidney cysts in *Cdh16^Cre^::Tfeb^fs^*, and *Cdh16^CreErt2^::Tfeb^fs^* mice. Number of cysts is shown as an average (± SEM) with bars sub-divided according to the dimension of the cysts. Cyst areas are presented as independent values (dots) with lines representing the means. Three-way (cyst number) and two-way (cyst area) Anova was applied. Cor, cortex; oMed, outer medulla; iMed, inner medulla. (**G**) Cadherin16 (CDH16) staining of kidneys from P30 mice. (**H**) Megalin, THP and AQP2 stainings in P90 *Cdh16^CreErt2^::Tfeb^fs^* mice. (**I**) PAS and Sirius Red staining. PAS staining shows the presence of single-layered or multi-layered cysts, and the presence of Clear Cells (CCs). SR staining shows areas of interstitial fibrosis, multi-layered basement membrane and protein casts. Asterisks, protein casts; sCy, simple Cysts; mCy, multilayered Cy; IF, Interstitial Fibrosis; mBM, multi-layered Basement Membrane. (*p<0.05, **p<0.01, ***p<0.001).

The following figure supplements are available for figure 1:

**Figure supplement 1.** Generation of transgenic mouse lines with kidney-specific *Tfeb* overexpression.

**Figure supplement 2.** Renal-specific *Tfeb* overexpression results in kidney enlargement and failure.

**Figure supplement 3.** Characterization of cyst origin in *Cdh16^Cre^::Tfeb^fs^* and *Cdh16^CreErt2^::Tfeb^fs^* mice.

the presence of very enlarged cells with a clear cytoplasm, which are commonly known as Clear Cells (CCs) (*Krishnan and Truong, 2002*) (*Figure 1I*). Sirius Red staining showed the presence of fibrosis and protein casts and revealed a significant accumulation of collagen inside the affected kidneys, as well as the presence of regions surrounded by multi-layered basement membranes (mBM) (*Figure 1I*). Importantly, the presence of Clear Cells, fibrosis and mBMs are characteristic features of kidneys from human patients with *TFEB*-fusion RCC (*Rao et al., 2012*).

## Identification of papillary renal cell carcinoma and of liver metastases

$^{18}$F-FDG PET analysis showed a higher glucose consumption in the kidneys of transgenic animals compared to controls, indicating a higher rate of glucose metabolism and suggesting a neoplastic transformation (*Figure 2A*). Similarly with PET analysis, HE and Ki67 stainings of the kidneys of *Cdh16^Cre^::Tfeb^fs^* mice revealed progressive hyperproliferation, which evolved into Ki67-positive neoplastic papillae at 5 months (*Figure 2B*). Neoplastic nodules, micropapillae and Hobnail-like cells, and mitotic spindles were detected at P12, 1 month, and 5 months, respectively (*Figure 2C–F*). Focal microcalcifications (*Figure 2G*), together with Clear Cells, and nests of neoplastic cells (*Figure 2H*) were also detected in *Cdh16^CreErt2^::Tfeb^fs^* mice.

Kidneys from both *Cdh16^Cre^::Tfeb^fs^* and *Cdh16^CreErt2^::Tfeb^fs^* mice presented numerous neoplastic lesions with both solid and cystic aspects, ranging from 0.102 to 2.93 mm and sometimes showing local invasion of the surrounding stroma (*Figure 2I*). Most importantly, liver metastases ranging from 0.9 to 3.8 mm, were found in both *Cdh16^Cre^::Tfeb^fs^* and *Cdh16^CreErt2^::Tfeb^fs^* mice. In *Cdh16^Cre^::Tfeb^fs^* animals, they were detected starting from P90 with an incidence of 23% (5 cases out of 21 *Cdh16^Cre^::Tfeb^fs^* mice older than 3 months). These metastases were positive for PAX8, that is a well-established marker for primary and metastatic RCC (*Ozcan et al., 2012*; *Shen et al., 2012*) and CDH16, which is a specific renal protein (*Shen et al., 2012*), while they were negative for the bile ducts and cholangiocarcinoma marker CK7 (Cytokeratin 7), consistent with their renal origin (*Figure 2L*).

## TFEB overexpression results in the induction of the canonical WNT pathway

To characterize the molecular mechanisms and identify the relevant pathways leading from *TFEB* overexpression to tumor development, we performed transcriptome analysis on kidney samples from *Cdh16^Cre^::Tfeb^fs^* and *Cdh16^Cre^* mice at P0 (GSE62977-KSP_P0 dataset) and at P14 (GSE63376-KSP_P14 dataset) (see Materials and methods) and found that *Tfeb* overexpression perturbed the kidney transcriptome in a statistically significant manner (*Figure 3—source data 1* and *2*, see also Materials and methods). Targeted analysis of the transcriptomic data revealed a significant induction of genes belonging to both ErbB and WNT signaling pathways. This was confirmed by real-time PCR

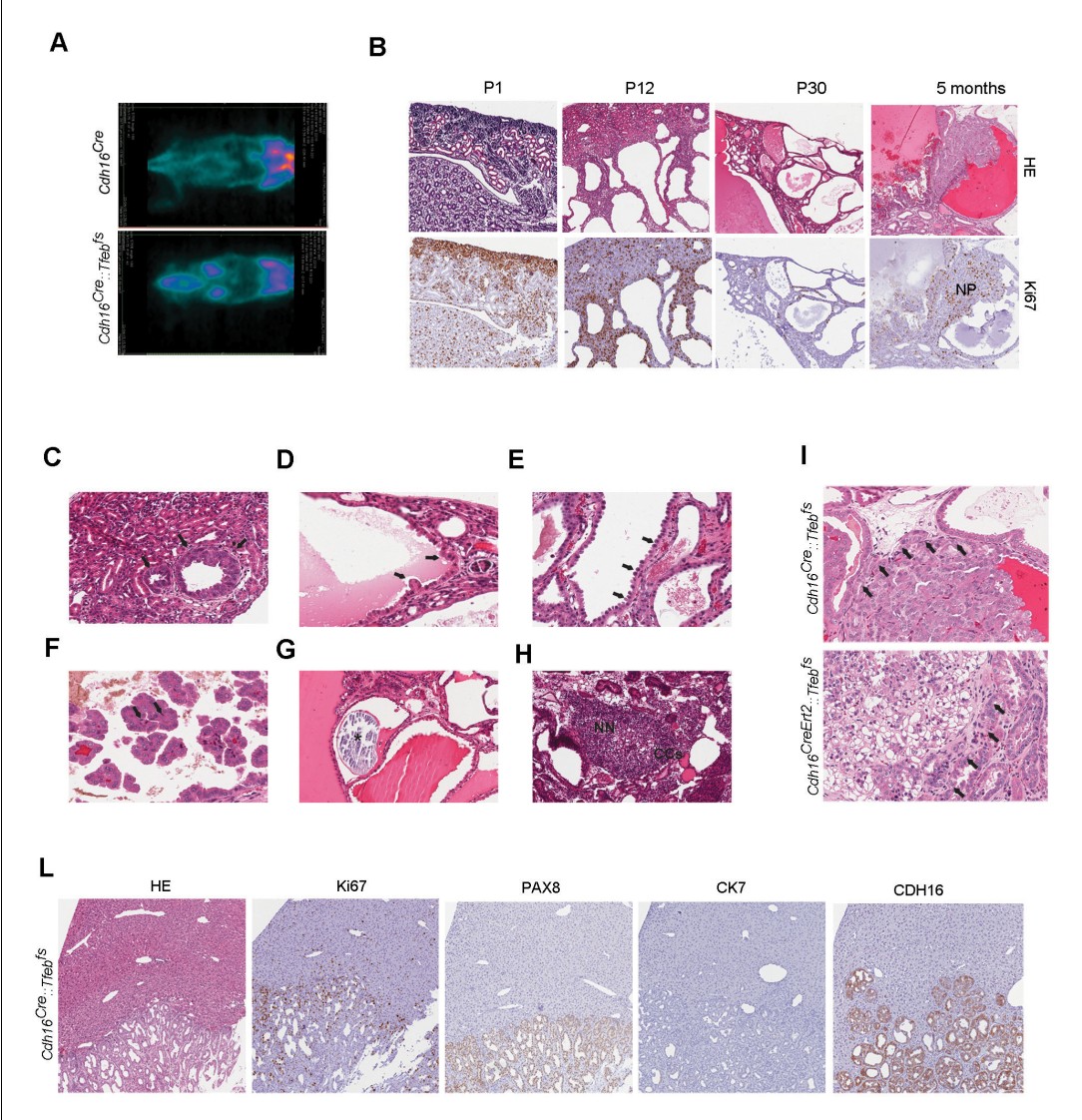

**Figure 2.** Kidney-specific Tfeb overexpression is associated with cancer development in *Cdh16^Cre^::Tfeb^fs^* and *Cdh16^CreErt2^::Tfeb^fs^* mice. (**A**) ^18^F-FDG PET/CT scan on P30 *Cdh16^Cre^::Tfeb^fs^* mice. (**B**) HE and Ki67 staining performed on *Cdh16^Cre^::Tfeb^fs^* mice at P1, P12, P30 and 5 months. Beginning at P12 the increase in cyst size is associated with an increase in papillary proliferation that becomes completely neoplastic by 5 months. NP, Neoplastic Papillae. (**C–H**) Representative images of neoplastic lesions at different stages: (**C**) neoplastic nodules (arrows) in P12 *Cdh16^Cre^::Tfeb^fs^* mice; (**D**) micropapillae (arrows) and (**E**) hobnail-like cells (arrows) in P30 *Cdh16^Cre^::Tfeb^fs^* mice; (**F**) mitotic spindles (arrows) in 5-month-old *Cdh16^Cre^::Tfeb^fs^* mice; (**G**) microcalcifications (asterisk) in tam-treated *Cdh16^CreErt2^::Tfeb^fs^* mice induced at P14 and sacrificed at 5 months; (**H**) neoplastic nests (NN) and clear cells (CCs) in tam-treated *Cdh16^CreErt2^::Tfeb^fs^* mice induced at P12 and sacrificed at P90. (**I**) HE staining of neoplastic lesions invading the surrounding stroma (arrows) in *Cdh16^Cre^::Tfeb^fs^* and in tam-treated *Cdh16^CreErt2^::Tfeb^fs^* mice. (**L**) Liver metastases in 5 month-old *Cdh16^Cre^::Tfeb^fs^* mice stained for HE, Ki67, PAX8 and CK7.

performed on *Cdh16^Cre^::Tfeb^fs^* mice at several developmental stages. Moreover, real-time PCR revealed an induction of *Myc* and *Axin2* genes, which are, together with *Ccnd1*, well-established WNT direct gene targets (*Clevers, 2006*) (*Figure 3A and B*, *Tables 1* and *2*). Kidneys from *Cdh16^CreErt2^::Tfeb^fs^* mice also had higher levels of all WNT-related genes that were identified in the constitutive line, and of many of the ErbB-related genes (*Figure 3—figure supplement 1A and B*).

Based on these results, we checked the activation of both ErbB and WNT signaling pathways. No evidence for an increase in the phosphorylation of AKT and ERK1/2 kinases (*Arteaga and Engelman, 2014*) was detected in P30 *Cdh16^Cre^::Tfeb^fs^* kidneys or in primary kidney cells obtained from

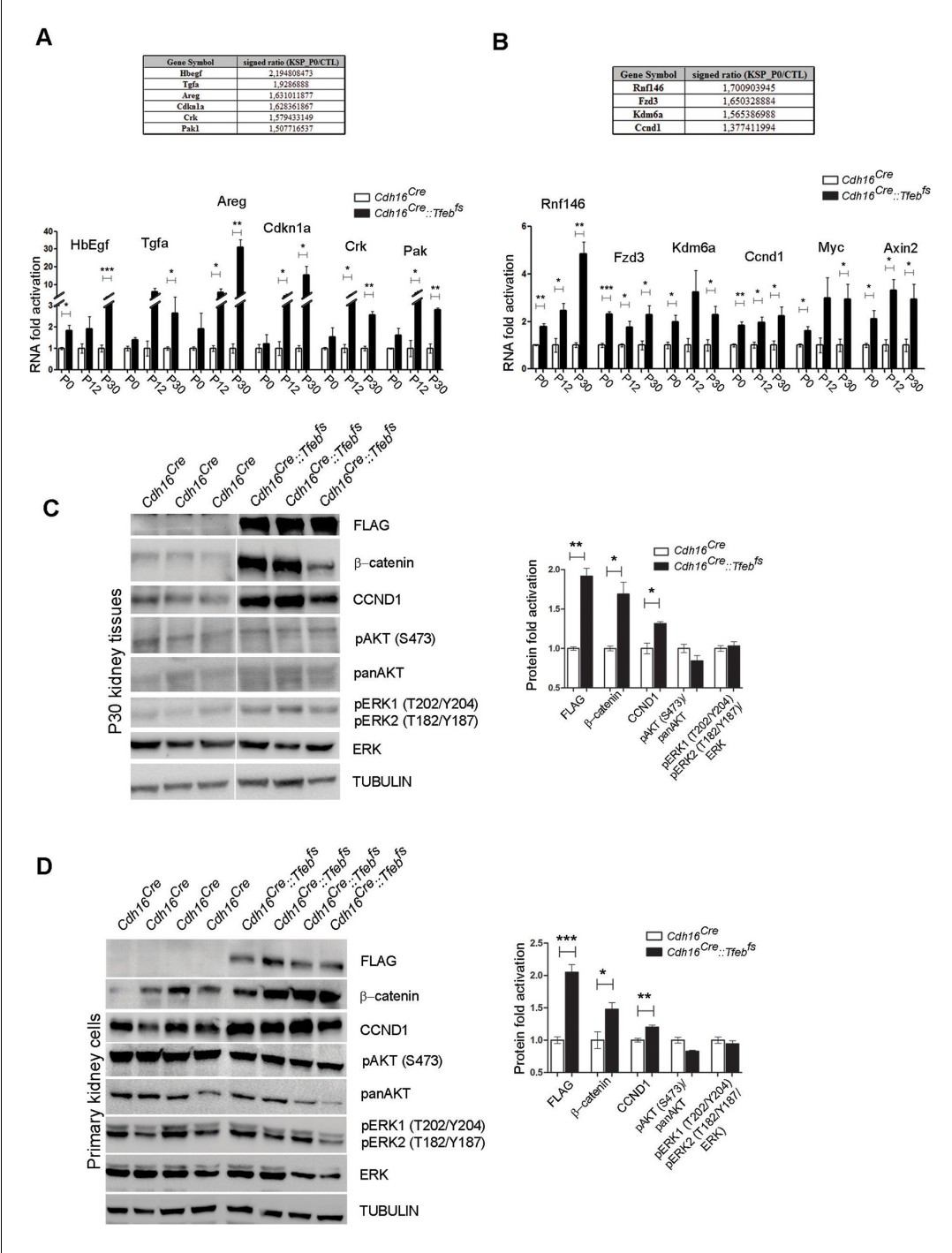

**Figure 3.** Activation of ErbB and WNT signaling pathways in kidneys from *Cdh16^Cre^::Tfeb^fs^* mice. Transcriptional and biochemical analyses were performed on *Cdh16^Cre^* and *Cdh16^Cre^::Tfeb^fs^* mice. (A,B) Tables show the relative increase of genes related to the ErbB (A) and WNT (B) pathways in the microarray analyses performed on kidneys from P0 *Cdh16^Cre^::Tfeb^fs^* mice. Graphs show real-time PCR validations performed on kidneys from *Cdh16^Cre^::Tfeb^fs^* mice at different stages (P0, P12, P30). Data are shown as the average (± SEM) of at least three *Cdh16^Cre^::Tfeb^fs^* mice normalized versus wild-type mice. (C,D) Immunoblot analyses performed on (C) P30 kidney tissues and (D) primary kidney cells isolated from *Cdh16^Cre^::Tfeb^fs^* mice to evaluate ErbB and WNT activation status. Each replicate is a distinct biological sample. ErbB signaling was assessed by looking at phosphoAKT (Ser473) to total AKT ratio, and phosphoERK1 (T202/Y204)/ERK2(T185/Y187) to total ERK ratio; WNT signaling was assessed by quantifying β-catenin and CCND1 (Cyclin D1) protein levels. Graphs represent the densitometry quantification of Western blot bands. Values are normalized to actin when not specified and are shown as an average (± SEM) (*p<0.05, **p<0.01, ***p<0.001, two-sided, Student's *t* test).

*Figure 3 continued on next page*

*Figure 3 continued*

The following source data and figure supplements are available for figure 3:

**Source data 1.** Complete list of 294 genes (represented by 361 probesets) significantly induced (FDR≤0.05) in the KSP_P0 microarray dataset (GSE62977).
**Source data 2.** Complete list of 628 genes (represented by 729 probesets) significantly induced (FDR≤0.05) in the KSP_P14 microarray dataset (GSE63376).
**Figure supplement 1.** ErbB and WNT transcriptional profiles in *Cdh16^CreErt2^::Tfeb^fs^* mice.
**Figure supplement 2.** Biochemical analysis of ErbB signaling.

transgenic mice (*Figure 3C and D*), indicating that the ErbB pathway was not induced. Erk1/2 activation, as detected by pERK1/2, was observed only at late stages (*Figure 3—figure supplement 2A*). The same result was observed in P14 and P30 tam-treated *Cdh16^CreErt2^::Tfeb^fs^* mice (*Figure 3—figure supplement 2B and C*). Conversely, we detected increased levels of total β-catenin and CCND1 in P30 renal tissues and primary kidney cells (*Figure 3C and D*) and increased levels of active β-catenin and of pLRP6 (Ser1490)/ LRP6 ratio in P30 and P90 renal tissues from *Cdh16^Cre^::Tfeb^fs^* mice (*Figure 4A and B*) and in P14 and P30 tam-treated *Cdh16^CreErt2^::Tfeb^fs^* mice (*Figure 4—figure supplement 1*). Moreover, β-catenin and active β-catenin staining of renal sections from *Cdh16^Cre^:: Tfeb^fs^* mice was significantly enhanced (*Figure 4C*). These results indicate the presence of a strong activation of the WNT signaling pathway in TFEB-overexpressing mice. Interestingly, the WNT pathway is known to play a role in renal cyst development (*Vainio and Uusitalo, 2000*; *Rodova et al., 2002*) and renal tumor formation, such as in VHL syndrome (*Peruzzi and Bottaro, 2006*) and Wilm's tumor (*Koesters et al., 1999*; *Zhu et al., 2000*; *Kim et al., 2000*). To investigate the role of TFEB in WNT pathway activation, we performed luciferase assays using a TOP-FLASH Luciferase WNT-reporter on immortalized kidney cell lines (HEK293 and HK2) co-transfected with *TFEB* and with both *β-catenin* and *TCF4* plasmids to stimulate WNT signaling. Luciferase activation was significantly higher in cells transfected with *TFEB* compared to controls without *TFEB*. No changes were observed when *TFEB* was transfected alone or only with *β-catenin* (*Figure 5A and B*). Together these data suggest that TFEB is able to enhance WNT pathway activation.

**Table 1.** ErbB-related genes up-regulated in the microarray analyses. (A) List of six genes with a known role in ErbB signaling pathway which are significantly up-regulated (FDR≤0.05) following TFEB overexpression in KSP_P0 microarray dataset (GSE62977). (B) One gene with a known role in ErbB signaling pathway which are significantly up-regulated (FDR≤0.05) following TFEB overexpression in KSP_P14 microarray dataset (GSE62977).

**A**

| Probe set ID | Gene symbol | Gene title | signed_ratio (KSP_P0/CTL) |
| --- | --- | --- | --- |
| 1418350_at | Hbegf | heparin-binding EGF-like growth factor | 2,194808473 |
| 1421943_at | Tgfa | transforming growth factor alpha | 1,9286888 |
| 1421134_at | Areg | amphiregulin | 1,631011877 |
| 1424638_at | Cdkn1a | cyclin-dependent kinase inhibitor 1A (P21) | 1,628361867 |
| 1425855_a_at | Crk | v-crk sarcoma virus CT10 oncogene homolog (avian) | 1,579433149 |
| 1450070_s_at | Pak1 | p21 protein (Cdc42/Rac)-activated kinase 1 | 1,507716537 |

**B**

| Probe set ID | Gene symbol | Gene title | signed_ratio (KSP_P14/CTL) |
| --- | --- | --- | --- |
| 1421134_at | Areg | amphiregulin | 1,221605795 |

**Table 2.** WNT-related genes up-regulated in the microarray analyses. (A) List of four genes with a known role in WNT signaling pathway which are significantly up-regulated (FDR≤0.05) following TFEB overexpression in KSP_P0 microarray dataset (GSE62977). (B) List of 10 genes with a known role in WNT signaling pathway which are significantly up-regulated (FDR≤0.05) following TFEB overexpression in KSP_P14 microarray dataset (GSE63376).

**A**

| Gene symbol | signed ratio (KSP_P0/CTL) |
| --- | --- |
| Rnf146 | 1,700903945 |
| Fzd3 | 1,650328884 |
| Kdm6a | 1,565386988 |
| Ccnd1 | 1,377411994 |

**B**

| Gene symbol | signed ratio (KSP_P14/CTL) |
| --- | --- |
| Rhou | 1,639718601 |
| Plcg2 | 1,601227563 |
| Gata3 | 1,358534898 |
| Fbxw2 | 1,262750602 |
| Mark2 | 1,248332335 |
| Axin1 | 1,21985179 |
| Tab1 | 1,217280695 |
| Psmb3 | 1,211737817 |
| Ndrg2 | 1,193338279 |
| Chd8 | 1,185904267 |

## Treatment with WNT inhibitors ameliorate the disease phenotype

Primary kidney cells derived from the renal cortex and medulla of $Cdh16^{Cre}::Tfeb^{fs}$ mice showed significantly higher levels of proliferation compared to wild-type cells (*Figure 5C*). We tested whether this hyperproliferative phenotype was sensitive to WNT inhibition. Strikingly, cell proliferation was significantly dampened, in a dose-dependent way, by two small-molecules, PKF118-310 and CGP049090 that specifically inhibit the WNT pathway by disrupting the interaction between β-catenin and TCF4 (*Avila et al., 2006*) and are known to suppress cell proliferation in several types of cancers, both in vitro and in vivo (*Wei et al., 2010*; *Wakita et al., 2001*) (*Figure 5D*). Moreover, β-catenin and CCND1 protein levels were highly reduced after PKF118-310 treatment (*Figure 5E*).

Based on the results obtained in primary kidney cells, we tested whether WNT inhibition could ameliorate the disease phenotype in vivo. P21 $Cdh16^{Cre}::Tfeb^{fs}$ transgenic animals were treated with daily IP injections of PKF118-310 for 30 days. At the end of the treatment, they showed an almost complete rescue of both cystic and cancer phenotypes (*Figure 6A*). Indeed, treated animals showed nearly normal KW/BW ratios (*Figure 6B*) and a significant reduction of many parameters of cystic and neoplastic pathology, such as the number and size of cysts and neoplastic papillae, and levels of Ki67 (*Figure 6C and D*, *Figure 6—figure supplement 1*, *Figure 6—source data 1*). We confirmed that drug-treatment in $Cdh16^{Cre}::Tfeb^{fs}$ mice suppressed the WNT pathway both at the mRNA and protein levels, as shown by the reduction of the mRNA levels of the WNT direct gene targets Cyclin D1, Myc and Axin2 (*Figure 6—figure supplement 2A*), by the reduction of Cyclin D1 and MYC proteins (*Figure 6—figure supplement 2B*) and by the decrease of Cyclin D1-positive nuclei in $Cdh16^{Cre}::Tfeb^{fs}$ drug-treated mice (*Figure 6—figure supplement 2C*). Furthermore, WNT inhibition resulted in normalization of expression levels of the gene encoding the transmembrane Glycoprotein nmb (*Gpnmb*) (*Figure 6E and F*), a known marker of melanomas, gliomas and breast cancers, which is also overexpressed in *TFE*-fusion ccRCCs (*Malouf et al., 2014*; *Zhou et al., 2012*). Interestingly,

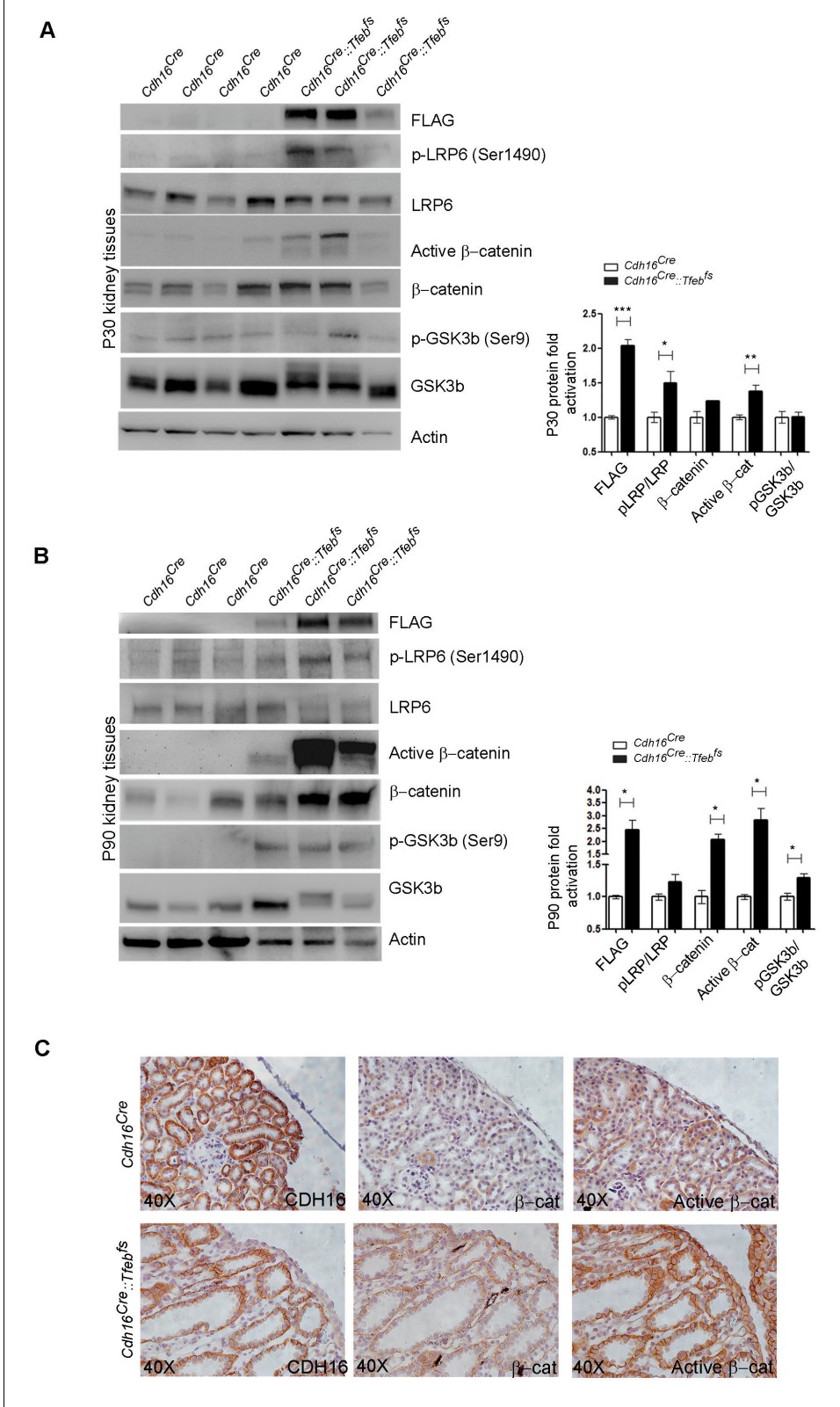

**Figure 4.** Molecular and histological analysis of WNT signaling. (**A,B**) Western blot analysis performed on (**A**) P30 and (**B**) P90 kidneys from *Cdh16*<sup>Cre</sup>::
*Tfeb*<sup>fs</sup> mice to assess WNT signaling activation by looking at different proteins related to this pathway. Each replicate is a distinct biological sample.
p-LRP6 (Ser1490)/LRP6, active β-catenin, β-catenin and p-GSK3β (Ser9)/GSK3β protein levels were quantified by densitometry analysis of the Western
blot bands. Values are normalized to actin when not specified, and are shown as an average (± SEM) (*p<0.05, **p<0.01, ***p<0.001, two-sided

*Figure 4 continued on next page*

*Figure 4 continued*

Student's *t* test). (C) Immunohistochemistry staining of CDH16, β-catenin and active β-catenin proteins performed on P30 kidney tissues from *Cdh16^Cre^:: Tfeb^fs^* mice.

The following figure supplement is available for figure 4:

**Figure supplement 1.** Molecular analysis of WNT signaling pathway in *Cdh16^CreErt2^::Tfeb^fs^* animals.

this gene is a direct target of TFEB, based on promoter (*Table 3*) and ChiP-Seq analysis (*Sardiello et al., 2009*) (*Table 4*).

## Autophagy is not required for disease progression

Considering the known role of TFEB as a master regulator of the lysosomal-autophagy pathway (*Argani et al., 2001*, *2005*; *Camparo et al., 2008*; *Davis et al., 2003*), and the recent evidence indicating that activation of autophagy driven by MiT/TFE genes plays an important role in pancreatic cancer (*Perera et al., 2015*), we tested whether autophagy plays a role in *TFE*-tRCC development. We analyzed the expression levels of a well-characterized panel of TFEB target genes known to be involved in lysosomal biogenesis and autophagy in *Cdh16^Cre^::Tfeb^fs^* mice. Surprisingly, no significant changes in the expression levels of these genes were detected in *Cdh16^Cre^::Tfeb^fs^* compared to wild type mice, with a few exceptions (*Figure 6—figure supplement 3A*). Consistently, immunoblot analysis of the autophagy marker LC3 in kidneys from transgenic mice did not reveal any significant changes compared to control littermates (*Figure 6—figure supplement 3B*). Furthermore, to test the role of autophagy in the pathogenesis of *TFE*-tRCC we crossed *Cdh16^Cre^::Tfeb^fs^* mice with autophagy-deficient *Atg7^flox/flox^* mice. No changes in kidney size or in the cystic phenotype were observed in TFEB overexpressing/autophagy-deficient double transgenic mice (*Atg7^flox/flox^:: Cdh16^Cre^::Tfeb^fs^*), herein referred to *Atg7^flox/flox^::Cdh16^Cre^::Tfeb^fs^*, compared to *Cdh16^Cre^:: Tfeb^fs^* mice (*Figure 6—figure supplement 3C–E*). Interestingly, most of the double transgenic animals died at approximately 1 month of age, suggesting that the combination of TFEB overexpression with autophagy inhibition in the kidney is toxic. This may be due to the previously described increase in sensitivity to oxidative stress of kidney-specific autophagy-deficient mice (*Liu et al., 2012*). These results suggest that autophagy does not play a critical role in the development of *TFE*-tRCC phenotype.

## Discussion

Kidney cancers associated with translocations of *TFE* genes represent a major unmet medical need (*Argani et al., 2005*; *Komai et al., 2009*; *Malouf et al., 2014*). Unfortunately, little is known about the mechanisms underlying this type of tumors.

In most cases, *TFEB*-tRCCs are associated to a well-characterized chromosomal translocation involving the TFEB gene and the non-coding Alpha gene, generating the alpha-*TFEB* fusion (t (6;11) (p21.2;q13) (*Davis et al., 2003*; *Kuiper et al., 2003*). Until recent reports, *TFEB* breakpoints were in all cases observed within a 289 bp cluster region (BCR) upstream exon 3, thus retaining the entire TFEB coding sequence (*Davis et al., 2003*; *Argani et al., 2005*; *Inamura et al., 2012*). As a consequence, the chromosomal translocation leads to a promoter substitution of the *TFEB* gene, and to a strong up-regulation of *TFEB* transcript and protein up to 60-times (*Kuiper et al., 2003*). Only recently, a new breakpoint was identified within exon 4, but the protein size appears to be the same as the wild-type protein (*Inamura et al., 2012*). In rare cases of RCCs, *TFEB* translocation partners were the *KHDBRS2* (*inv(6)* (*p21;q11*)) (*Malouf et al., 2014*) and the *CLTC* (t(6;17) (*p21;q23*)) genes (*Durinck et al., 2015*). The situation of *TFE3* chromosomal translocations appears to be more complicated. *TFE3* was found to be involved in translocations with five known gene partners (i.e. *PRCC, ASPSCR1, SFQP, NONO, CLTC*) leading to the generation of fusion proteins. The identification of multiple *TFE3*-gene partners and the characterization of two *TFE3*-fusion proteins (*TFE3-NONO, TFE3-SFQP*) (*Clark et al., 1997*) strongly suggested that RCC is caused by *TFE3*, rather than by its partners (*Kauffman et al., 2014*). Indeed, *TFE3* fusion protein resulted to be much more stable and transcriptionally active than the wild-type protein (*Weterman et al., 2000*). Together, these

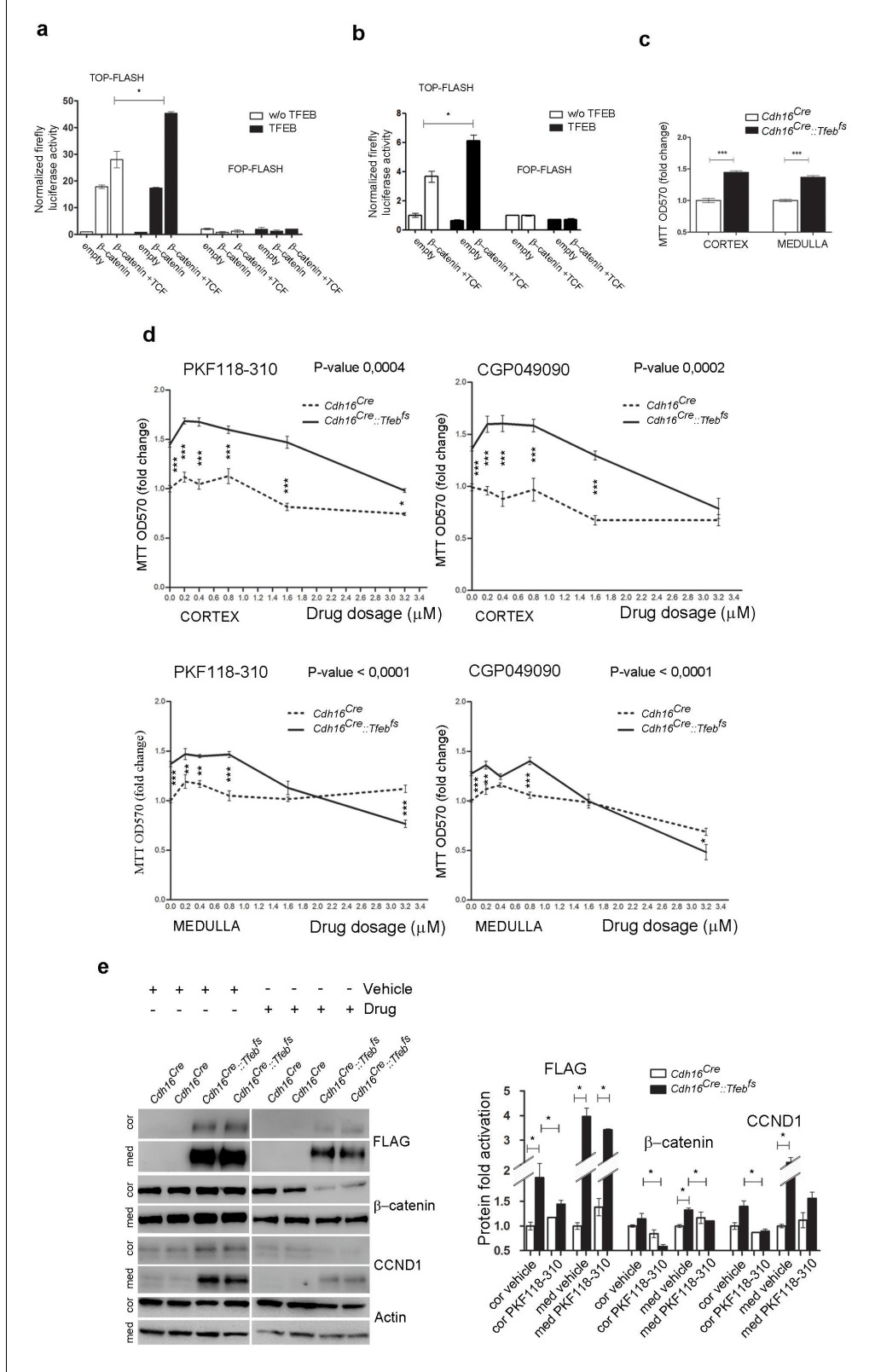

**Figure 5.** Inhibition of WNT signaling rescues the hyper-proliferative phenotype of kidney cells from *Cdh16/Tfeb* mice. (A,B) Activity of the TCF/LEF reporter *TOP-FLASH*. Luciferase activity after co-transfection of *β-catenin* and *TCF* plasmids in HEK293 (A) and HK2 (B) cells with and without *Tfeb* overexpression. Values are shown as an average (± SEM) of each point in duplicate, normalized to the Renilla values and to the basal condition. Data are representative of three independent experiments. (C) MTT tetrazolium reduction assay (MTT) was used to evaluate proliferation of primary kidney

*Figure 5 continued on next page*

Figure 5 continued

cells derived from *Cdh16^Cre^::Tfeb^fs^* mice. Values are shown as an average (± SEM) of each point in triplicate and normalized versus wild-type mice. Data are representative of three independent experiments. (D) MTT proliferation assays of primary kidney cells treated independently with two WNT signaling inhibitors, PKF118-310 and CGP049090, added at different dosages for 24 hr. 0 µm represents the basal proliferation of cells. Values are shown as means (± SEM) of three replicates per point normalized to the vehicle (DMSO), added at the same concentration, and versus the *Cdh16^Cre^* cells without drug treatment. Results are representative of three independent experiments. Two-way Anova was applied (factors: cell genotype, treatment). (E) Immunoblot analysis on primary kidney cells treated with Drug (PKF118-310) or Vehicle (DMSO) for 24 hr at 1.6 µM. Graphs show the densitometry quantifications of Western blot bands. Values are normalized to actin and are shown as averages (± SEM) (Cor, cortex; Med, medulla). (*$p<0.05$, **$p<0.01$, ***$p<0.001$).

data suggest that the first step, and driving force, of the disease pathological cascade is the overexpression of active TFEB and TFE3 proteins, which is likely associated to an increase of their function as transcription factors.

Currently, there are no model systems to study the mechanisms underlying *TFE*-tRCC kidney tumors and to identify and test new therapeutic strategies. Until now, very limited data were available on the biological pathways involved in these tumors. Argani et al. (*Argani et al., 2010*) reported activation of the mTOR pathway in *TFE*-tRCC patients compared to ccRCCs, as shown by increased phosphorylation levels of the downstream mTOR target S6. Unfortunately, selective mTORC1 inhibition performed on patients with *TFE*-tRCCs did not improve the disease phenotype (*Malouf et al., 2010*). Up-regulation of the MET-tyrosine kinase receptor, which in turn activates HGF-signaling, was detected in *TFE*-tRCC patients by in vitro assays (*Tsuda et al., 2007*), but subsequent analyses on TFE3-renal samples failed to identify activated MET protein (*Kauffman et al., 2014*). The lack of mechanistic insights in *TFE*-t*RCCs* have hampered the identification of effective therapeutic strategies (*Kauffman et al., 2014*). Some patients with metastatic *TFE3*-tRCC have been treated with inhibitors of ErbB receptors and of the mTOR pathway. Unfortunately, most of these patients relapsed after an initial period of remission (*Parikh et al., 2009*; *Wu et al., 2008*).

The lack of knowledge of the mechanisms underlying *TFE*-t*RCCs* prompted us to generate transgenic mouse models that overexpress *TFEB* in the kidney, thus mimicking the human disease situation. We generated two transgenic mouse models overexpressing *TFEB* in the epithelial cells of the kidney in either a constitutive (*Cdh16^Cre^::Tfeb^fs^*) or an inducible (*Cdh16^CreErt2^::Tfeb^fs^*) manner. A severe renal cystic pathology associated with a significant increase in renal size was observed in these mice. In the constitutive model, cysts arose from the collecting ducts and distal tubules, whereas in the inducible one they derived from proximal and distal tubules.

We observed that cysts were either single- or multi-layered. Epithelial cells lining the mono-layered cysts often lost their cuboidal shape, becoming flattened. Further analyses revealed the presence of protein casts inside the cysts and multi-layered basal membranes in the regions surrounding the cysts, due to collagen deposition. Interestingly, the presence of fibrosis, mBMs and tubular or cystic structures covered by a single layer of flattened, cuboidal, and columnar cells is also observed in human patients affected by *TFEB*-tRCCs (*Rao et al., 2012*, *2013*). Finally, in both types of transgenic lines, we observed the presence of highly enlarged cells with a clear cytoplasm, that closely resemble the 'Clear Cells' found in human patients with RCC (*Rao et al., 2012*).

Transgenic mice also displayed a higher glucose metabolism, as shown by PET-scan performed in P30 animals suggesting the presence of renal cancer. At P12, *Cdh16^Cre^::Tfeb^fs^* mice already presented cystic changes together with neoplastic nodules that were Ki67-positive. The progressive hyper-proliferation of these nodules resulted in the development of micropapillae starting from P30, which evolved into neoplastic papillae in 5-month-old mice. Finally, liver metastases positive for PAX8 and CDH16 and neoplastic nests were observed in older animals. These data indicate that these newly generated transgenic lines bear all major histological and phenotypic features of human *TFE*-tRCC (*Kauffman et al., 2014*; *Rao et al., 2012*, *2013*), thus representing excellent models to study this disease.

To identify the effect of TFEB overexpression on the kidney transcriptome, we performed microarray analysis on kidney samples from P0 *Cdh16^Cre^::Tfeb^fs^* mice. Unexpectedly, transgenic mice did not show a significant induction of the autophagy machinery and crossing of these animals with an

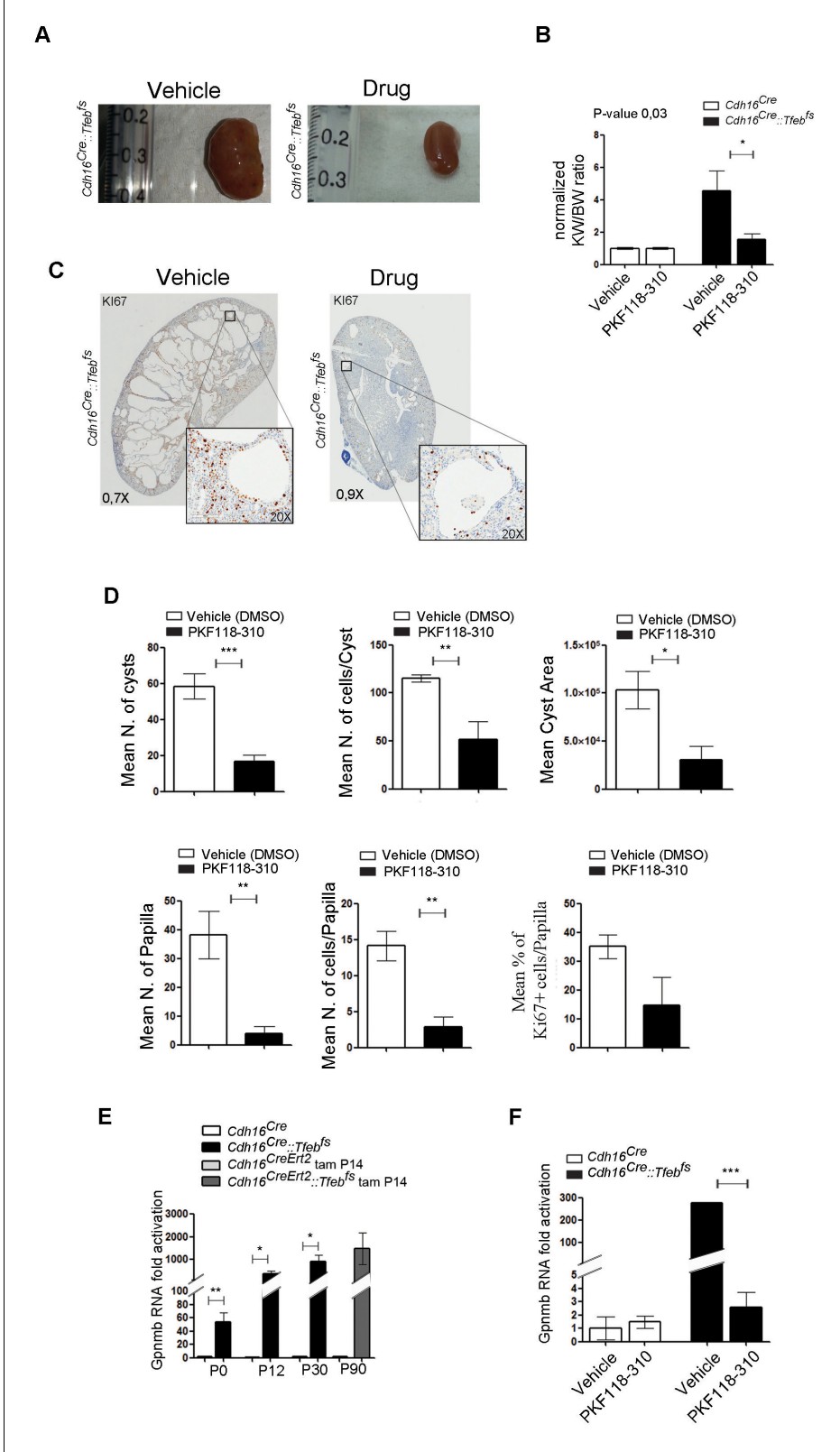

**Figure 6.** Treatment with WNT inhibitor attenuates cystic and neoplastic phenotypes. Morphological, histological and biochemical analyses performed on *Cdh16^Cre^::Tfeb^fs^* mice treated with Vehicle (DMSO) or Drug (PKF118-310). (**A,B**) Kidney images (**A**) and sizes (KW/BW) (**B**) from *Cdh16^Cre^::Tfeb^fs^* mice injected intraperitoneally (IP) either with vehicle or drug at 0.85 mg/kg. KW/BW ratios are shown as means (± SEM) and values are normalized to the *Cdh16^Cre^* animals treated with vehicle. Two-way ANOVA was applied (factors: treatment, genotype). (**C**) Ki67 staining of kidneys from *Cdh16^Cre^::Tfeb^fs^*

*Figure 6 continued on next page*

*Figure 6 continued*

mice after treatment with vehicle or drug. Insets are enlargements of a single cyst. (D) Quantification of several parameters related to cysts and papillae performed on kidney sections from vehicle- and PKF118-310-treated *Cdh16^Cre^::Tfeb^fs^* mice. (E) Gpnmb mRNA fold activation in kidneys from *Cdh16^Cre^:: Tfeb^fs^* and tam-treated *Cdh16^CreErt2^::Tfeb^fs^* mice at different stages. Values are shown as means (± SEM) of at least three mice and each group is normalized to the proper control (respectively *Cdh16^Cre^* and tam-treated *Cdh16^CreErt2^*). (F) Gpnmb fold activation in kidneys from *Cdh16^Cre^::Tfeb^fs^* mice treated with vehicle or PKF118-310. Values are shown as means (± SEM) of at least three animals per group and are all normalized versus the *Cdh16^Cre^* mice treated with vehicle. (*p<0.05, **p<0.01, ***p<0.001, two-sided Student's *t* test).

The following source data and figure supplements are available for figure 6:

**Source data 1.** Numerical data of each parameter showed in *Figure 6D* and divided per genotype and treatment.

**Figure supplement 1.** In vivo treatment of *Cdh16^Cre^::Tfeb^fs^* mice with the WNT inhibitor PKF118-310 partially rescues cystic and neoplastic phenotypes.

**Figure supplement 2.** In vivo treatment of *Cdh16^Cre^::Tfeb^fs^* mice with the PKF118-310 drug inhibits WNT pathway overactivation.

**Figure supplement 3.** Inhibition of autophagy in *Tfeb* overexpressing mice (*Atg7^flox/flox^::Cdh16^Cre^::Tfeb^fs^*) does not affect the cystic phenotype.

autophagy deficient *Atg7^flox/flox^* mouse line failed to revert the disease phenotype, thus suggesting that autophagy does not play an essential role in the pathogenesis of this disease.

Transcriptome analysis revealed a significant induction of genes involved in the WNT pathway, such as WNT direct target genes *Ccnd1*, *Myc* and *Axin2* and WNT-related genes *Fzd3*, *Rnf146* and *Kdm6a*. This transcriptional induction was consistent with increased protein levels of total β-catenin, active β-catenin, CCND1 and pLRP6 (Ser1490)/ LRP6 ratio. Furthermore, an induction of the phospho-GSK3β (Ser9)/ GSK3β ratio, an inactive form of the GSK3β kinase, was detected at later stages. Hyper-activation of the WNT pathway was also observed in cortical and medullary primary kidney cells derived from *Cdh16^Cre^::Tfeb^fs^* mice. Most importantly, luciferase assays performed on HEK-293 and HK-2 cells revealed that TFEB overexpression resulted in a significant enhancement of WNT pathway activation.

WNT signalling is of central importance for the development of many organs and has been implicated in tumor pathogenesis at different sites such as skin (*Robbins et al., 1996*), brain (*Zurawel et al., 1998*), liver (*de La Coste et al., 1998*) and prostate (*Voeller et al., 1998*). Its activation requires the formation of the WNT signalosome, resulting from the binding of WNT ligands to Frizzled (Fzd) receptors. This mediates the interaction of Fzd with LRP5/6 proteins. Fzd-LRP5/6 hetero-oligomerization is required to sequester the β-catenin degradation complex, containing several kinases such as GSK3 and CK1. GSK3 is then able to phosphorylate LRP but not β-catenin. Active β-catenin translocates into the nucleus and activates its target genes, such as *MYC*, *AXIN2* and

**Table 3.** GPNMB expression profiles and CLEAR sites. (A) Differentially expression of Gpnmb transcript in KSP_P0 (GSE62977), in KSP_P14 microarray dataset (GSE63376) and in RCC dataset. (B) Sequence analysis of the CLEAR sites (i.e. the consensus TFEB binding sites) in the human and murine promoter region of Gpnmb.

**A**

| Probe set ID | Gene symbol | Gene title | Representative public ID | Ensembl | ratio (KSP_P0/ CTL) | ratio (KSP_P14/ CTL) | ratio (RCC/ CTL) |
|---|---|---|---|---|---|---|---|
| 1448303_at | Gpnmb | glycoprotein (transmembrane) nmb | NM_053110 | ENSMUSG00000029816 | 10,61358979 | 4,926015853 | 141,4101213 |

**B**

| Gene | Score | Sequence | Chrom | ABS start | ABS end | TSS_position |
|---|---|---|---|---|---|---|
| *Gpnmb* | 0,8731563 | GGGGCAAGTGACTC | chr6 | 49036518 | 49036531 | 1 |
| *Gpnmb* | 0,803943 | ACATCACATGATCT | chr6 | 49036587 | 49036600 | 70 |
| *GPNMB* | 0,8484716 | CCATCACATGATCC | chr7 | 23286328 | 23286341 | 13 |

**Table 4.** List of 11 genes shared between the KSP_P0 dataset and from an HeLa TFEB-overexpressing ChIP-Seq dataset.

| Gene symbol | signed_ratio (KSP_P0/CTL) | chromosome | start | stop | peak tags | distance from 5' end of gene | RefSeq ID | symbol | ID | ABS distance |
|---|---|---|---|---|---|---|---|---|---|---|
| Elf3 | 1,881188134 | chr1 | 2E+08 | 201978977 | 8 | -712 | NM_001114309 | ELF3 | ETS-related transcription factor Elf-3 | 712 |
| Gna13 | 1,504591673 | chr17 | 6E+07 | 63053379 | 8 | -58 | NM_006572 | GNA13 | guanine nucleotide-binding protein subunit | 58 |
| Ankrd12 | 1,599217835 | chr18 | 9E+06 | 9137025 | 15 | 0 | NM_015208 | ANKRD12 | ankyrin repeat domain 12 isoform 1 | 0 |
| Atp6v1c1 | 1,658752808 | chr8 | 1E+08 | 104033525 | 15 | 0 | NM_001695 | ATP6V1C1 | V-type proton ATPase subunit C 1 | 0 |
| Bhlhe40 | 2,03490115 | chr3 | 5E+06 | 5021164 | 10 | 0 | NM_003670 | BHLHE40 | class E basic helix-loop-helix protein 40 | 0 |
| Gpnmb | 10,61358979 | chr7 | 2E+07 | 23286524 | 9 | 0 | NM_002510 | GPNMB | transmembrane glycoprotein NMB isoform b | 0 |
| Kdm6a | 1,58385317 | chrX | 4E+07 | 44732628 | 33 | 0 | NM_021140 | KDM6A | lysine-specific demethylase 6A | 0 |
| Lats2 | 1,761917857 | chr13 | 2E+07 | 21636098 | 22 | 0 | NM_014572 | LATS2 | serine/threonine-protein kinase LATS2 | 0 |
| Ppargc1a | 2,713649997 | chr4 | 2E+07 | 23891989 | 11 | 0 | NM_013261 | PPARGC1A | peroxisome proliferator-activated receptor gamma | 0 |
| Rnf146 | 1,700903945 | chr6 | 1E+08 | 127588198 | 14 | 0 | NM_030963 | RNF146 | ring finger protein 146 | 0 |
| Usp2 | 2,284889961 | chr11 | 1E+08 | 119252760 | 8 | 0 | NM_004205 | USP2 | ubiquitin specific peptidase 2 isoform a | 0 |

CCND1 (*Clevers, 2006*), by interacting with the TCF4/LEF1 transcription factors (*Voronkov and Krauss, 2013*).

Interestingly, hyper-activation of the WNT pathway was recently detected in a melanoma cell line in which MITF, another member of the MiT/TFE family, was overexpressed, leading to an expansion of the endo-lysosomal compartment that in turn was able to concentrate and relocate the WNT signalosome/destruction complex and consequently to enhance WNT signaling (*Ploper et al., 2015*). In addition, several studies have linked alterations in the regulation of the β-catenin pathway to abnormalities of kidney development and function (*Vainio and Uusitalo, 2000*). Indeed, β-catenin is necessary for proper regulation of the *PKD1* promoter (*Rodova et al., 2002*), that is mutated in 85% of patients with Autosomal Dominant Polycystic Kidney Disease (ADPKD). Furthermore, the WNT pathway is also known to play a role in renal tumor formation, such as in VHL syndrome (*Peruzzi and Bottaro, 2006*) and Wilm's tumor (*Koesters et al., 1999*; *Zhu et al., 2000*; *Kim et al., 2000*). Mice lacking the *Apc* gene specifically in the kidney are prone to the development of cystic renal cell carcinomas (*Sansom et al., 2005*). Finally, cytoplasmic accumulation of β-catenin was observed in patients with *TFE3*-tRCC, suggesting the presence of a possible link between *TFE*-factors and WNT-signaling components (*Bruder et al., 2007*). Together these studies reveal a strong link between hyper-activation of WNT signaling and tumorigenesis in the kidney and reinforce our finding of WNT hyper-activation in TFEB transgenic mice as a critical step of the disease pathogenesis.

Based on this evidence, we postulated that treatment with WNT inhibitors had beneficial effects on *TFE*-tRCCs. To test this hypothesis, we treated primary kidney cells from *Cdh16^Cre^::Tfeb^fs^* mice with two small molecules, PKF118-310 and CGP049090, able to inhibit the WNT pathway by disrupting the interaction between β-catenin and TCF-4 (*Avila et al., 2006*). Drug treatments significantly reduced the hyper-proliferation rate observed in cells from transgenic mice, bringing it to normal levels. Therefore, we sought to reproduce these data in vivo by treating *Cdh16^Cre^::Tfeb^fs^* mice with WNT inhibitors. Administration of the PKF118-310 molecule or vehicle for 30 days resulted in a substantial reduction of several important parameters, such as kidney size, cyst number and size, Ki67 index and the number of neoplastic papillae. Moreover, drug-treated *Cdh16^Cre^::Tfeb^fs^* animals

showed a significant decrease in the mRNA levels of Gpnmb, a known marker of melanomas, gliomas and breast cancer, which was reported to be overexpressed in *TFE*-fusion *ccRCCs* (*Malouf et al., 2014*; *Zhou et al., 2014*). Interestingly, we also found that Gpnmb is a direct transcriptional target of TFEB (*Sardiello et al., 2009*).

This study provides direct evidence that overexpression of *TFEB* in the kidney is able to generate a severe cystic pathology associated with the development of kidney cancer and liver metastases, thus mimicking the cancer phenotype associated with human *TFE*-fusion *ccRCCs* chromosomal translocations. Thus, the transgenic mouse lines that we generated represent the first genetic animal models of renal cell carcinoma. The study of these mice revealed that WNT activation plays a crucial role in *TFE-tRCCs* and that WNT inhibitors can be used to rescue the phenotype of our transgenic mouse models, suggesting that targeting WNT signaling could be a promising therapeutic approach for the treatment of *TFE-tRCC* patients.

## Materials and methods

### Mouse models

*Tfeb*^fs/fs^ transgenic mice (generated by Dr. Settembre [*Settembre et al., 2011*]) were crossed with a kidney-specific *Cdh16*^Cre^ (*Cdh16*, Cadherin 16) (Jackson laboratories RRID:IMSR_JAX:012237) and *Cdh16*^CreErt2^ (generated by Dr. Peters [*Lantinga-van Leeuwen et al., 2006*]) mice. The *Atg7* conditional KO mice (*Komatsu et al., 2005*) (*Atg*^flox/flox^ mice) was a generous gift from T.Eissa. Mice were crossed with *Cdh16*^Cre^and *Tfeb*^fs/fs^ mice to obtain kidney-specific Atg7 deletion and TFEB overexpression (*Atg*^flox/flox^: :Cdh16^Cre^::Tfeb^fs^). All mice used were maintained in a C57BL/6 background genotype. *Cdh16*^Cre^and *Cdh16*^Cre^::Tfeb^fs^ mice were injected intra-peritoneally (IP) with tamoxifen at a dosage of 100 µg/g of mouse weight for three consecutive days to obtain an efficient recombination. For the Kidney to Body weight ratio experiments, we analyzed at least three animals per genotype/sex/condition, but often the number was higher than 5. Experiments were conducted in accordance with the guidelines of the Animal Care and Use Committee of Cardarelli Hospital in Naples and authorized by the Italian Ministry of Health.

### Cell culture, transfections and plasmids

Primary kidney cells were obtained following the protocol described in Leemans et al. (*Leemans et al., 2005*). Briefly, kidneys were collected and uncapsulated. Tissue from the outer cortex and inner medulla was cut into approximately 1 mm³ pieces, and subsequently digested by 1 mg/ml collagenase type 1A (Sigma- Aldrich, Saint Louis, MO) at 37°C for 1 hr. After washing cells with PBS, primary TECs were grown to confluence in DMEM-F12 culture medium supplemented with 10% FCS, 100 IU/ml penicillin, 100 mg/ml streptomycin, 2 mM L-glutamine (Gibco; Invitrogen Corp.), 1% ITSe and 1% S1 hormone mixture (Sigma-Aldrich) and were cultured in 5% CO₂ at 37 degrees. TECs were identified by characteristic cobblestone-shaped morphology. Tfeb overexpression was confirmed by FLAG immunoblot (*Figure 2*). HEK293 (CRL-1573, RRID:CVCL_0045) and HK2 (CRL-2190, RRID:CVCL_0302) cells were purchased from ATCC. The identity of these cells have been confirmed by STR profiling (http://web.expasy.org/cellosaurus/CVCL_00459) (http://web.expasy.org/cellosaurus/CVCL_0302). No mycoplasma contamination was detected in these cells. HEK293 cells were cultured in DMEM (Euroclone) supplemented with 10% FBS, 100IU/ml penicillin, 100 mg/ml streptomycin and 2 mM L-glutamine (Gibco; Invitrogen Corp.). HK2 cells were grown in DMEM-F12 (Invitrogen) supplemented with 5% FBS, 100 IU/ml penicillin, 100 mg/ml streptomycin, 2 mM L-glutamine (Gibco; Invitrogen Corp.) and 1% ITSe. Cells were grown at 5% CO₂ at 37 degrees. Human full-length TFEB-FLAG was previously described (*Settembre et al., 2011*). The Top-Flash and FopFlash plasmids (Upstate), the pCS2+MT-Myc-tagged β-CATENIN (full-length β-CATENIN), and the Evr2-Tcf1E plasmid (Tcf1E) were kindly provided by Dr. M. Plateroti.

Cells were transfected with Lipofectamine LTX and Plus reagent (Invitrogen) following the manufacturer's protocol. Luciferase activity was measured 48 hr post-transfection using the Dual-Luciferase Reporter Assay System (Promega). To normalize transfection efficiency in reporter assays, the HEK293 and HK2 cells were co-transfected with a plasmid carrying the internal control reporter *Renilla reniformis* luciferase driven by a TK promoter (pRL-TK; Promega). Data are representative of

three independent experiments and statistical significance was determined using Student's *t*-test. p<0.05 was considered as significant.

## In vitro drug treatments and MTT proliferation assay

Cultured primary kidney cells derived from the cortex and medulla of $Cdh16^{Cre}$ and $Cdh16^{Cre}::Tfeb^{fs}$ mice were seeded in 96-well plates at the density of $5 \times 10^3$ cells/well, maintained overnight at 37°C, and incubated in the presence of the test compounds at the different concentrations. PKF118-310 and CGP049090 were added at different dosages (0 μm, 0.2 μm, 0.4 μm, 0.8 μm, 1.6 μm, 3.2 μm) for 24 hr. 0 μm represents the basal proliferation of cells after 48 hr of plating. MTT assay was used to assess cell proliferation. Briefly, 5 mg of MTT powder was solubilized in 1 mL of PBS and filtered. Ten microliter of this solution was added to 100 μl of cell culture medium without phenol red. At the end of the incubation time, cells were washed twice with PBS and incubated with MTT-media solution to form formazan crystals. After 4 hr, media was removed and 100 μl/well of a solubilisation solution was added to the cells (2.1 mL HCl 10 N, 500 mL isopropanol) for 4 hr at 37°C to obtain a complete solubilization of the crystals. As a readout, absorbance of the 96-well plate was measured recording the Optical Density (OD) at 570 nm with a microplate spectrophotometer system. Results are representative of three independent experiments performed on three different $Cdh16^{Cre}$ and $Cdh16^{Cre}::Tfeb^{fs}$ mice. *T*-test is referred to cells without drug (0 μm) taken from $Cdh16^{Cre}::Tfeb^{fs}$ mice versus cells without drug (0 μm) taken from $Cdh16^{Cre}$ mice. Data are representative of three independent experiments, and statistical significance was determined using Student's *t*-test. p<0.05 was considered as significant.

## In vivo drug treatments

P21 $Cdh16^{Cre}$ and $Cdh16^{Cre}::Tfeb^{fs}$ mice were injected IP daily, from Monday to Friday, with the PKF118-310 drug at the dose of 0.85 mg/kg or with an equal amount of vehicle (DMSO). After 30 days from the beginning of the treatment, animals were sacrificed and kidneys were collected and weighted and processed for further analyses. Six animals for each group and genotype were collected.

## Biochemical analysis

Plasma urea was measured using standardized clinical diagnostic protocols of the Academical Medical Center Amsterdam. Albumin (Bethyl Laboratories, Montgomery, TX) was measured in urines collected for 24 hr in metabolic cages and was analyzed by following the manufacturer's instructions.

## High-frequency ultrasound and PET/CT scan analyses

All the imaging procedures were performed with mice under general anesthesia. Anesthesia was produced in an induction chamber, saturated with 5% isoflurane (Iso-Vet 1000 mg/g Inhalation Vapor, Piramal Healthcare UK Ltd., Northumberland, UK) in oxygen (2 L/min) and subsequently maintained during all procedures with a conenose delivering isoflurane at 1.5% in oxygen at 2 L/min.

For High-frequency ultrasound, each mouse was placed in dorsal recumbency on a dedicated, heated, small animal table (VEVO Imaging Station 2, FUJIFILM VisualSonics, Inc., Toronto, Ontario, Canada) and hairs were removed with a small clipper and then with the application of a depilatory cream, and a pre-warmed ultrasound-coupling gel was applied to the skin to improve ultrasound transmission and reduce contact artefacts. A 40 MHz transducer (MS 550 D, FUJIFILM VisualSonics, Inc., Toronto, Ontario, Canada) was mounted on the dedicated stand of the imaging station, and B-mode and Color-Doppler mode images were obtained on the ultrasound equipment (VEVO 2100, FUJIFILM VisualSonics, Inc., Toronto, Ontario, Canada).

Positron emission tomography (PET) coupled with computed tomography (CT) was performed with a dedicated small animals PET/CT scanner (eXplore Vista, GE Healthcare), with a trans-axial field of view of 6.7 cm and an axial field of view of 4.8 cm. Animals, fasted overnight, were injected under general anesthesia in the lateral caudal vein with 300 μCi of [$^{18}$F]-fluorodeoxyglucose (FDG). Mice were left to recover from anesthesia under a heating lamp and PET/CT acquisitions were started after 90 min of biodistribution. Static emission scans of 30 min with energy window of 250–700 keV were acquired. The PET datasets were reconstructed by 2D FORE/3D OSEM algorithm and corrected for random coincidences, scatter, physical decay to the time of injection (voxel size: 0.3875 ×

0.3875 × 0.775 mm³). The mean specific uptake value (SUV) was obtained for each region of interest using the visualization and analysis software of the scanner (version 4.11 Build 701, MMWKS Image Software: Laboratorio de Imagen, HGUGM, Madrid, Spain).

## Survival analysis

Survival curves were calculated for a period of 8 months on a total of 15 $Cdh16^{Cre}::Tfeb^{fs}$ mice, 10 $Cdh16^{CreErt2}::Tfeb^{fs}$ mice (tam P12), 12 $Cdh16^{CreErt2}$ mice (tam P14) and 12 $Cdh16^{CreErt2}::Tfeb^{fs}$ mice (tam P30) grown in the same animal facility, all in same background (C57BL/6). Values were plotted by the product-limit method of Kaplan and Meier; statistical analyses were carried out applying the Log Rank (Mantel-Cox) test.

## Quantitative real-time PCR

Total RNA was isolated from frozen samples lysed in Trizol (Life Technologies) using a TissueLyser (Qiagen) and following the recommended manufacturer's protocol. Reverse transcription was performed using QuantiTect Rev Transcription Kit (Qiagen). Finally, real-time PCR was performed using SYBR Green (Roche Diagnostics) and performing the reaction in the LightCycler System 2.0 (Roche Applied Science). The parameters of real-time PCR amplification were defined according to Roche recommendations. To quantify gene expression, Gapdh mRNA expression was used as an internal reference. All the values are shown as fold activation respect to w-type levels. Data are representative of three independent experiments and statistical significance was determined using Student's *t*-test. p<0.05 was considered as significant.

The following primers were used in this study: Gapdh; forward (fw) tgcaccaccaactgcttagc, reverse (rev) tcttctgggtggcagtgatg; Tfeb; fw gcagaagaaagacaatcacaa, rev gccttggggatcagcatt; Ccnd1; fw ccttgactgccgagaagttgtg, rev gttccacttgagcttgttcacca; Axin2; fw gatgcatcgcagtgtgaagg, rev ggttccacaggcgtcatctc; Myc; fw ccagcagcgactctgaagaa, rev acctcttggcaggggtttg; Fzd3; fw gcatctgggagacaacatgg, rev caggtctggacgactcatctg; Rnf146; fw agcggaggagaaaagactgc, rev acatagccctttctcggtccg; Kdm6a; fw tgacagcggaggagagggag, rev ccttcatcctggcgccatct; Cdkn1a; fw gtctgagcggcctgaagatt, rev caatctgcgcttggagtgat; HbEgf; fw tccacaaaccagctgctacc, rev ccttgtggcttggaggagaa; Pak1; fw ttcctgaaccgctgtcttga, rev tcaggctagagaggggcttg; Areg; fw tattggcatcggcatcgtta, rev tgcacagtcccgtttcttg; Crk; fw cgcgtctcccactacatcat, rev tctcctattcggagcctgga; Tgfa; fw agtgcccagattcccacact, rev cgtacccagagtggcagaca; Gpnmb, fw tggctacttcagagccacca, rev ggcatggggacatctgctat.

## Microarray hybridization

Total RNA (3 µg) was reverse transcribed to single-stranded cDNA with a special oligo (dT) 24 primer containing a T7 RNA promoter site, added 3′ to the poly-T tract, prior to second strand synthesis (One Cycle cDNA Synthesis Kit by Affymetrix, Fremont, CA). Biotinylated cRNAs were then generated, using the GeneChip IVT Labeling Kit (Affymetrix). Twenty microgram of biotinylated cRNA was fragmented and 10 µg hybridized to the Affymetrix GeneChip Mouse 430A_2 microarrays for 16 hr at 45°C using an Affymetrix GeneChip Fluidics Station 450 according to the manufacturer's standard protocols.

For the analysis at P0, the total RNA was extracted from the kidney of three $Cdh16^{Cre}::Tfeb^{fs}$ mice and of two control $Cdh16^{Cre}$ mice. For the analysis at P14, total RNA was extracted from the kidney of three $Cdh16^{Cre}::Tfeb^{fs}$ P14 mice and three control $Cdh16^{Cre}$ P14 mice.

## Microarray data processing

The data discussed in this publication have been deposited in NCBIs Gene Expression Omnibus (GEO) (*Edgar et al., 2002*) and are accessible through GEO Series accession number GSE62977 (KSP_P0 dataset) and GSE63376 (KSP_P14 dataset) (KSP, Kidney specific). Low-level analysis to convert probe level data to gene level expression was performed using Robust Multiarray Average (RMA) implemented using the RMA function of the Bioconductor project (*Gentleman et al., 2004*).

## Statistical analysis of differential gene expression

For each gene, a Bayesian t-test (Cyber-t) (*Baldi and Long, 2001*) was used on RNA normalized data to determine if there was a significant difference in expression between $Cdh16^{Cre}::Tfeb^{fs}$ mice versus

*Cdh16^{Cre}* mice both at P0 (GSE62977-KSP_P0 dataset) and at P14 (GSE63376- KSP_P14 dataset). p-Value adjustment for multiple comparisons was done with the False Discovery Rate (FDR) of Benjamini-Hochberg (*Klipper-Aurbach et al., 1995*). The threshold for statistical significance chosen was FDR≤0.05. In the KSP_P0 dataset, we selected 361 probe-sets corresponding to 294 significantly induced genes (GSE62977). In the KSP_P14 dataset, we selected 729 probe-set corresponding to 628 genes (GSE63376).

## (Immuno-) histological analysis

Formalin-fixed, paraffin-embedded kidney sections (4 μm) were analyzed using standard hematoxylin and eosin (HE) staining, periodic acid Schiff (PAS) staining, or Sirius Red (SR) staining. For immunohistochemistry procedures, sections were subjected to heat-mediated antigen retrieval procedure (10 mM citrate buffer pH 6.0) followed by 1 hr preincubation with normal goat serum (1:200; Dako-Cytomation, Glostrup, Denmark). After blocking of endogenous peroxidase activity for 15 min in 0.1% $H_2O_2$ in water, sections were incubated with primary antibodies diluted in 1% BSA in PBS. Following incubation with secondary antibody, immune reactions were revealed using NovaRed or diaminobenzidine chromogen and counterstained with hematoxylin, dehydrated, and mounted.

Primary antibodies: rabbit polyclonal anti-megalin (1:750, Pathology LUMC, Leiden, the Netherlands), goat polyclonal anti-uromodulin (1:4000, Organon Teknika-Cappel,Turnhout, Belgium), rabbit polyclonal anti-aquaporin-2 (1:4000 Calbiochem, Amsterdam, The Netherlands), rabbit polyclonal anti-β-catenin (1:500, Santa Cruz sc-7199, RRID:AB_634603), rabbit monoclonal anti-active β-catenin (1:800, Cell Signaling #8814, RRID:AB_11127203), rabbit polyclonal anti-cadherin16 (1:300, Novus NBP159248, RRID:AB_11046440), rabbit polyclonal anti-ATG7 (1:300, Santa Cruz sc-33211, RRID: AB_2062165), rabbit monoclonal anti-Ki67 (ABCAM ab16667, clone SP6, RRID:AB_302459, 1:200), a rabbit polyclonal anti-PAX8 antibody (Proteintech, 10336-1-AP, RRID:AB_2236705, 1:1000) and a mouse monoclonal anti-Cytokeratin 7 (Abcam, ab9021, RRID:AB_306947, 1:500). Secondary antibodies: anti-rabbit envision HRP (DakoCytomation, Glostrup, Denmark), rabbit-anti-goat HRP (1:100), power rabbit poly-HRP (Biocare Medical, M4U534L). For staining with Sirius Red, de-paraffinized sections were incubated in 0.2% phosphomolybdic acid hydrate for 5 min and 0.1% Sirius red for 90 min. Subsequently, sections were incubated for 1 min in saturated picric acid and then placed in 70% ethanol, dehydrated and mounted.

## Quantitative histology

Histomorphometric analysis were conducted on PAS and Ki67-stained sections. For the cyst characterization, cyst number and area was calculated on PAS sections from three animals per genotype and group. Cysts were hand-annotated and measured in the outer and inner cortex, and the outer and inner medulla. Finally, they were sub-divided according to their size.

For the analyses performed on the drug- and vehicle-treated animals, the analysis was conducted on Ki67-stained sections. The number and size of the cysts were defined within the areas identified by the pathologist using ImageScope (Leica-Biosystems Nussloch GmbH).

Using the same method, the number of papillae was counted and the proportion of Ki-67 positive nuclei on the total number of nuclei within the papillae was calculated. For these analyses, a total of six *Cdh16^{Cre}::Tfeb^{fs}* vehicle (DMSO)-treated and six *Cdh16^{Cre}::Tfeb^{fs}* drug (PKF118-310)-treated animals were evaluated.

## Antibodies and western blotting

Tissues were microdissected and disrupted using a TissueLyser (Qiagen). Cells or tissues were lysed by solubilisation in lysis buffer (50 mM Tris at pH 7.9, 1% Triton X-100, 0,1% Tween 20, 150 mM NaCl, 5 mMMgCl2, 10% glycerol) containing phosphatase (Roche) and protease (Sigma) inhibitors. Protein concentration was measured by the Bradford method. Samples were mixed with Laemmli lysis buffer, boiled and resolved by SDS-PAGE. Thereafter, proteins were blotted onto polyvinylidene fluoride (PVDF) membranes and blocked for 1 hr with non-fat 5% milk or 5% BSA diluted in 1X TBS, 0,1% Tween 20, according to the primary antibody protocol. Membranes were incubated with primary antibodies overnight. Visualization was made by incubation with corresponding HRP-labeled secondary antibodies (Calbiochem) followed by enhanced chemiluminescence (ECL) (Perkin Elmer, Waltham, MA). Membranes were developed using a Chemidoc UVP imaging system (Ultra-Violet

Products Ltd) and densitometric quantification was performed in unsaturated images using ImageJ (NIH).

For Western blots, the following antibodies were used: anti-FLAG M2-HRP (Sigma, cat. A8592, RRID:AB_439702, 1:1000), anti-actin (Sigma, cat. A2066, RRID:AB_476693, 1:5000), anti-βtubulin (Sigma, cat. T8328, RRID:AB_1844090 1:1000), anti-Human/Mouse/Rat Pan-Akt (R&D, cat. MAB2055, RRID:AB_2224581, 1:500), Phospho-Akt (Ser473) (D9E) Cell Signaling, cat. #4060, RRID: AB_2315049, 1:1000), anti-human, mouse, and rat ERK1/ERK2 (R&D, cat.216703, RRID:AB_2140121, 1:2000), anti-Human/Mouse/Rat Phospho- ERK1(T202/Y204)/ERK2 (T185/Y187) (R&D, cat. AF1018, RRID:AB_354539 1:1000), anti-β-catenin (BD, cat. 610154, RRID:AB_397555 1:500), anti-active β-catenin (Cell Signaling, cat. #8814, RRID:AB_11127203 1:1000), anti-Cyclin D1 (Cell Signaling, cat. #2978, RRID:AB_10692801 1:1000), anti-LRP6 (Cell Signaling, cat. #3395, RRID:AB_1950408 1:1000), anti-phospho-LRP6 (Ser1490) (Cell Signaling, cat. #2568, RRID:AB_2139327 1:1000), anti-GSK3β (Cell Signaling, cat. #9315, RRID:AB_490890 1:1000), anti-phospho-GSK3β (Ser9) (Cell Signaling, cat. #9323, RRID:AB_2115201 1:1000), anti MYC (Cell Signaling, cat. #5605, RRID:AB_1903938 1:1000).

## Statistical analysis

GraphPad Prism (GraphPad Software, San Diego, CA) was used for all statistical analysis. Statistical analyses of data were performed using Student's t-test. One-way ANOVA and Tukey's post-hoc tests were performed when comparing more than two groups relative to a single factor (time or treatment/genotype). Two-way and three-way ANOVA and Tukey's post-hoc tests were performed when comparing more than two groups relative to two or more factors. Mantel-Cox test was used for the survival analysis. $p < 0.05$ was considered significant.

## Acknowledgements

We thank A De Matteis, D Bagley and G Diez-Roux for critical reading of the manuscript, L D'Orsi, D Ricca, C Luise, G Jodice and A C Salzano for technical support, R Andolfi for animal handling, A Carissimo for statistical analysis, L Auletta for imaging analyses and M Plateroti for the *β-catenin* and *TCF4* plasmids. This work was supported by grants from the Italian Telethon Foundation (TGM11CB6), the European Research Council Advanced Investigator grant no. 250154 (CLEAR) (AB); US National Institutes of Health (R01-NS078072) (AB), the Associazione Italiana per la Ricerca sul Cancro (AIRC) (AB) (IG 2015 Id 17639), the Associazione Italiana per la Ricerca sul Cancro (AIRC - IG 11904 to SP; 14404 to PPDF; and MCO 10.000 to PPDF and SP), MIUR (the Italian Ministry of University and Scientific Research), the Italian Ministry of Health to SP and PPDF and the Monzino Foundation to PPDF.

## Additional information

### Funding

| Funder | Grant reference number | Author |
|---|---|---|
| Fondazione Telethon | TGM11CB6 | Andrea Ballabio |
| European Research Council | 250154 | Andrea Ballabio |
| National Institutes of Health | R01-NS078072 | Andrea Ballabio |
| Associazione Italiana per la Ricerca sul Cancro | IG 2015 Id 17639 | Andrea Ballabio |
| Associazione Italiana per la Ricerca sul Cancro | IG 11904 | Salvatore Pece |
| Associazione Italiana per la Ricerca sul Cancro | 14404 | Pier Paolo Di Fiore |
| Associazione Italiana per la Ricerca sul Cancro | MCO 10.000 | Salvatore Pece Pier Paolo Di Fiore |
| Ministero della Salute | | Salvatore Pece Pier Paolo Di Fiore |

| Fondazione Antonio Carlo Monzino | Pier Paolo Di Fiore |

The funders had no role in study design, data collection and interpretation, or the decision to submit the work for publication.

## Author contributions

AC, PPDF, Conception and design, Acquisition of data, Analysis and interpretation of data, Drafting or revising the article; Lkors, EV, RDC, NZ, EN, SC, GB, SP, GGM, EdH, MS, Conception and design, Acquisition of data, Analysis and interpretation of data; CS, JCL, Conception and design, Acquisition of data, Analysis and interpretation of data, Contributed unpublished essential data or reagents; DJMP, AB, Conception and design, Acquisition of data, Analysis and interpretation of data, Drafting or revising the article, Contributed unpublished essential data or reagents

## Author ORCIDs

Andrea Ballabio, http://orcid.org/0000-0003-1381-4604

## Ethics

Animal experimentation: Experiments were conducted in accordance with the guidelines of the Animal Care and Use Committee of Cardarelli Hospital in Naples and authorized by the Italian Ministry of Health, approved protocol number: 75/2014-B.

## Additional files

### Major datasets

The following datasets were generated:

| Author(s) | Year | Dataset title | Dataset URL | Database, license, and accessibility information |
| --- | --- | --- | --- | --- |
| Rossella De Cegli | 2016 | Expression data from mice overexpressing Tcfeb specifically in P14 kidney | http://www.ncbi.nlm.nih.gov/geo/query/acc.cgi?acc=GSE63376 | Publicly available at the NCBI Gene Expression Omnibus (accession no: GSE63376) |
| Rossella De Cegli | 2016 | Expression data from mice overexpressing Tcfeb specifically in P0 kidney | http://www.ncbi.nlm.nih.gov/geo/query/acc.cgi?acc=GSE62977 | Publicly available at the NCBI Gene Expression Omnibus (accession no: GSE62977) |

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
