## [Decision Letter]

Thank you for submitting your article "Modelling TFE renal cell carcinoma in mice reveals a critical role of WNT signaling" for consideration by *eLife*. Your article has been reviewed by three peer reviewers, one of whom is a member of our Board of Reviewing Editors and the evaluation has been overseen by a Reviewing Editor and Sean Morrison as the Senior Editor. The reviewers have opted to remain anonymous.

The reviewers have discussed the reviews with one another and the Reviewing Editor has drafted this decision to help you prepare a revised submission.

Summary:

The authors describe a new mouse model of TFEB overexpression in the kidney to mimic renal tumors with translocation-induced gain of function of TFE transcription factors. TFEB overexpression results in cyst formation and neoplastic lesions in the kidney. Gene expression studies revealed dramatic induction of genes in the WNT signaling pathway in the affected kidneys, and inhibitors of this pathway significantly ameliorated disease in mice.

Essential revisions:

1) The mouse model should be described in more detail, particularly the prevalence and size of tumor nodules, whether the primary tumors display local invasion, and the incidence of metastasis to the liver.

2) The presence of apparent liver metastases from the renal tumors is significant but requires validation. Can the authors perform molecular or other analyses to confirm that the liver lesions arise from tumors in the kidney?

3) Figure 6: Please provide evidence by IHC or PCR that the WNT pathway has been suppressed in the drug-treated kidneys.

4) It is surprising that, based on expression analysis of a few markers, cysts appear to develop from different cell types in the inducible and non-inducible lines. The authors should provide an explanation for this unusual phenomenon, and ideally use additional markers to strengthen their conclusion.

---

## [Author Response]

*1) The mouse model should be described in more detail, particularly the prevalence and size of tumor nodules, whether the primary tumors display local invasion, and the incidence of metastasis to the liver.*

We observed neoplastic nodules, starting from P12, in all *KSPcdh16/Tcfeb* and *indKSPcdh16/Tcfeb* mice (on a total of 43 analysed mice), with a size ranging from 0.102 to 2.93 mm and sometimes presenting local invasion of the surrounding stroma as shown by new Figure 2. We identified liver metastases ranging from 0.9 to 3.8 mm, in both *KSPcdh16/Tcfeb* and *indKSPcdh16/Tcfeb* mice. Five mice out of a total of 21 *KSPcdh16/Tcfeb* mice analysed over three months of age (an incidence of 23%) developed liver metastases. This information has been included in the text.

*2) The presence of apparent liver metastases from the renal tumors is significant but requires validation. Can the authors perform molecular or other analyses to confirm that the liver lesions arise from tumors in the kidney?*

As already shown in Figure 2, liver metastases were positive for PAX8, that is a well-established marker for primary and metastatic RCC (Ozcan A. *et al.*, Arch Pathol Lab Med. 2012; Shen SS. *et al.*, Arch Pathol Lab Med. 2012) and for *KSP*-CDH16, which is a specific renal protein, that is not expressed in other tissues (Shen SS. *et al.*, Arch Pathol Lab Med. 2012). Indeed *Ksp*-Cdh16 promoter has been used to drive renal CRE-specific expression in this study and in a number of other studies (Shao X., *J Am Soc Nephrol.* 2002; Kai T., *J Pathol.* 2016). Liver metastases were instead negative for the bile ducts and cholangiocarcinoma marker CK7 (Cytokeratin 7), consistent with their renal origin (Figure 2). This information has been included in the text. We also show that the *KSP*-CDH16 protein was detectable together with the FLAG on liver metastases by Western blot, while healthy hepathic tissue did not show any *KSP*-CDH16 protein expression (see figure below).

Author response image 1.**DOI:**
http://dx.doi.org/10.7554/eLife.17047.024

3) Figure 6: Please provide evidence by IHC or PCR that the WNT pathway has been suppressed in the drug-treated kidneys.

As suggested by the reviewer, to check if the WNT pathway was suppressed in drug-treated animals we quantified mRNA levels of the well-established WNT direct gene targets CyclinD1, c-Myc and Axin2 (Clevers 2006), which were significantly induced in the *KSPcdh16/Tcfeb* mouse model. We first validated their induction in vehicle-treated *KSPcdh16/Tcfeb* mice and subsequently demonstrated their suppression in drug-treated *KSPcdh16/Tcfeb* mice. Moreover, biochemical analysis showed a reduction of CyclinD1 and c-MYC proteins in drug-treated mice. Finally, IHC analysis showed a significant decrease of CyclinD1-positive nuclei in drug-treated mice. These data have been included in Figure 6—figure supplement 2.

*4) It is surprising that, based on expression analysis of a few markers, cysts appear to develop from different cell types in the inducible and non-inducible lines. The authors should provide an explanation for this unusual phenomenon, and ideally use additional markers to strengthen their conclusion.*

We agree with the reviewer’s observation and have now included a discussion of this issue in the text. Previous studies on mouse models of polycystic kidney disease have shown that cysts may originate from different cell types (Latinga Van Leeuwen IS. *et al.*, Hum Mol Genet. 2007; Happé H. *et al.*, Hum Mol Genet. 2009; Leonhard WN. *et al.*, J Am Soc Nephrol. 2016). These differences can be explained by different sensitivities to stimuli of different renal segments at different developmental stages (Latinga Van Leeuwen IS. *et al.*, Hum Mol Genet. 2007; Happé H. *et al.*, Hum Mol Genet. 2009; Leonhard WN. *et al.*, J Am Soc Nephrol. 2016; Piontek K. *et al.*, Nat Med. 2007). As part of a separate ongoing, and still unpublished study, Dr. Peters (co-author of our paper) crossbred iKsp-Pkd1del mice with LacZ reporter mice to visualize the pattern of Cre activity upon Tamoxifen administration. She showed that Tamoxifen at either P10,11,12 or at P18,19,20 results in a similar pattern of Cre activity, whereas the cystic phenotype is quite different. Notably, Pkd1 inactivation at P10 rapidly leads to severe cyst formation specifically in the region with the least Ksp-Cre activity, suggesting that this region of the nephron is especially sensitive to cyst formation during a developmental stage. By contrast, Pkd1 deficient cells from the inner medulla, rarely develop cysts. These experiments confirmed the previous indication that different renal segments have different sensitivities and that these vary over time.